# Synthesis and Hydrogen Production Performance of MoP/a-TiO_2_/Co-ZnIn_2_S_4_ Flower-like Composite Photocatalysts

**DOI:** 10.3390/molecules28114350

**Published:** 2023-05-25

**Authors:** Keliang Wu, Yuhang Shang, Huazhen Li, Pengcheng Wu, Shuyi Li, Hongyong Ye, Fanqiang Jian, Junfang Zhu, Dongmei Yang, Bingke Li, Xiaofei Wang

**Affiliations:** 1Henan Key Laboratory of Microbial Fermentation, School of Biology and Chemical Engineering, Nanyang Institute of Technology, Nanyang 473000, Chinayehongyong2016@163.com (H.Y.);; 2Anhui Yansheng New Material Co., Ltd., Hefei 230039, China; 3School of Chemistry and Chemical Engineering, Shihezi University, Shihezi 832003, China; 4Department of Petroleum and Chemical, Bayingoleng Vocational and Technical College, Bazhou 841000, China; 5College of Materials and Chemical Engineering, West Anhui University, Lu’an 237012, China

**Keywords:** ZnIn_2_S_4_, Co doping, TiO_2_, photocatalysis, hydrogen evolution

## Abstract

Semiconductor photocatalysis is an effective strategy for solving the problems of increasing energy demand and environmental pollution. ZnIn_2_S_4_-based semiconductor photocatalyst materials have attracted much attention in the field of photocatalysis due to their suitable energy band structure, stable chemical properties, and good visible light responsiveness. In this study, ZnIn_2_S_4_ catalysts were modified by metal ion doping, the construction of heterojunctions, and co-catalyst loading to successfully prepare composite photocatalysts. The Co-ZnIn_2_S_4_ catalyst synthesized by Co doping and ultrasonic exfoliation exhibited a broader absorption band edge. Next, an a-TiO_2_/Co-ZnIn_2_S_4_ composite photocatalyst was successfully prepared by coating partly amorphous TiO_2_ on the surface of Co-ZnIn_2_S_4_, and the effect of varying the TiO_2_ loading time on photocatalytic performance was investigated. Finally, MoP was loaded as a co-catalyst to increase the hydrogen production efficiency and reaction activity of the catalyst. The absorption edge of MoP/a-TiO_2_/Co-ZnIn_2_S_4_ was widened from 480 nm to about 518 nm, and the specific surface area increased from 41.29 m^2^/g to 53.25 m^2^/g. The hydrogen production performance of this composite catalyst was investigated using a simulated light photocatalytic hydrogen production test system, and the rate of hydrogen production by MoP/a-TiO_2_/Co-ZnIn_2_S_4_ was found to be 2.96 mmol·h^−1^·g^−1^, which was three times that of the pure ZnIn_2_S_4_ (0.98 mmol·h^−1^·g^−1^). After use in three cycles, the hydrogen production only decreased by 5%, indicating that it has good cycle stability.

## 1. Introduction

The energy crisis is an ongoing global issue of increasing importance. Moreover, the rapid development of industrialization around the world has led to severe energy and environmental pressures [1]. Thus, there is an increased emphasis on research worldwide to successfully address the global energy crisis and to create new sustainable sources of energy [2]. The capture and conversion of solar energy by the photocatalytic splitting of water offers a promising strategy for converting inexhaustible solar energy into hydrogen (H_2_) energy [3]. However, there are currently two main constraints that limit the large-scale application of hydrogen: (1) the large-scale green synthesis of hydrogen is a significant challenge; (2) the storage and transport of hydrogen is also difficult [4]. Hydrogen’s shortcomings are partly explained by high infrastructure costs for production, storage, and distribution. These problems may result from their low energy density per volume, explosive characteristics, and ability to cause embrittlement in metals such as steel [5]. Many methods have been investigated for the production of hydrogen. The photocatalytic decomposition of water for hydrogen production is one of the simplest, most environmentally friendly and low-cost methods for producing hydrogen. Therefore, this method has attracted extensive research attention [6]. In particular, the production of hydrogen via the solar photolysis of water is gaining increasing attention due to its potential for solving the global energy crisis and mitigating environmental pollution problems [7,8]. Photogenerated charge carriers can be excited from photocatalysts under sunlight, and after the photogenerated electrons migrate to the surface of semiconductors, H^+^ in water receives electrons that are reduced to H_2_. The holes left behind are combined with sacrificial agents in the system and used to achieve continuous H_2_ production.

In photocatalytic systems, the mobility of photogenerated carriers is an important factor affecting photocatalytic efficiency, with a fast migration rate and high separation efficiency positively contributing to the photocatalytic reaction [9]. The electrostatic potential of ZnIn_2_S_4_ with a hexagonal laminar structure is uniformly distributed within the plane, and the small potential of this material is well conducive to carrier migration [10]. Moreover, the positive charges are densely distributed in the indium sulfide tetrahedra and octahedra within the cell, while the negative charges are concentrated in the zinc indium tetrahedral [11]. Therefore, photogenerated electrons are easily transferred to the indium sulfide polyhedra, while the photogenerated holes more easily migrate to the zinc indium tetrahedra, which improves the separation efficiency of the photogenerated carriers [12]. Furthermore, the band gap of ZnIn_2_S_4_ is 2.3~2.5 eV and the energy band of ZnIn_2_S_4_ is narrow, which is also conducive to the generation of photogenerated carriers [13]. ZnIn_2_S_4_ is therefore an ideal photocatalytic material with broad application prospects.

In 2003, Lei et al. [14] synthesized ZnIn_2_S_4_ by a hydrothermal method and used this material as an effective visible-light-driven hydrogen precipitation photocatalyst for the first time. Guoʹs group [15] synthesized ZnIn_2_S_4_ microspheres by a hydrothermal/solvothermal process and explored their visible-light-driven photocatalytic hydrogen production performance. Their findings showed that these microsphere catalysts had a good potential for producing photocatalytic hydrogen from water when exposed to visible light [16].

However, pure ZnIn_2_S_4_ photocatalysts still suffer from low visible light utilization and low photocatalytic activity [17]. Moreover, the photocatalytic activity of ZnIn_2_S_4_ semiconductors is affected to some extent by their limited photogenerated electron and hole separation efficiency under visible light irradiation and low photogenerated carrier mobility [18]. Therefore, Yuan Wenhui et al. [19] prepared a series of Co-doped ZnIn_2_S_4_ photocatalysts using a solvothermal synthesis method. The successful incorporation of Co into the ZnIn_2_S_4_ lattice was confirmed by X-ray diffraction (XRD) and X-ray photoelectron spectroscopy (XPS). With increasing Co concentration, the absorption edge of the samples caused red-shift, but the Co also gradually disrupted the ZnIn_2_S_4_ morphology. Their photocatalytic results showed that Co^2+^ doping significantly improved the photocatalytic activity of ZnIn_2_S_4_. The optimum Co doping amount of 0.3 wt% for the ZnIn_2_S_4_ photocatalyst led to the highest photocatalytic activity [20]. Therefore, in this work, a doping amount of 0.3% was chosen to preserve the petal-like morphology and enhance the specific surface area of ZnIn_2_S_4_ while also improving its hydrogen production performance and utilization of sunlight [21,22].

TiO_2_ has been widely investigated as a semiconductor photocatalyst material due to its many advantages, such as high stability and high photosensitivity. Therefore, TiO_2_-based metal oxide photocatalysts are widely used in many practical applications [23]. However, TiO_2_ particles easily agglomerate, have a low adsorption capacity for organic matter, and exhibit low solar energy utilization [24]. These factors limit the photocatalytic efficiency of TiO_2_ and seriously affect its application in practical production [25]. The focus of photocatalytic research has therefore shifted from the improvement of traditional TiO_2_ performance to the investigation of other catalysts with better performance in the visible light range. Amorphous TiO_2_ is an important category of TiO_2_ materials that exhibits the common “short-range order, long-range disorder” [26] structural feature seen in amorphous materials. Amorphous semiconductors have a large number of suspended bonds. Therefore, the energy band structures of amorphous materials exhibit a gap band between the valence band and the conduction band [27]. Amorphous TiO_2_ with a lower band gap width can be obtained by modifying its electronic structure. This reduces the energy intensity required for electrons to transfer from the valence band to the conduction band [28]. Therefore, visible light irradiation can be used to activate these materials, improving their photocatalytic activity [29]. Zywitzki et al. reported amorphous titania-based photocatalysts synthesized using a facile, UV-light mediated method and evaluated as photocatalysts for hydrogen evolution from water/methanol mixtures. The resulting amorphous materials exhibited an overall higher hydrogen evolution rate (1.09 mmol·h^−1^·g^−1^) compared to a crystalline TiO_2_ reference (P25 0.80 mmol·h^−1^·g^−1^) on a molar basis of the photocatalyst due to their highly porous structure and high surface area [30].

The photocatalytic activity of a photocatalyst is determined by its light absorption capacity as well as its electron–hole transfer and separation efficiency [31]. These factors are related to the catalyst surface properties, which play an important role in photocatalytic processes. For instance, the loading of co-catalysts on a photocatalyst surface to provide hydrogen production sites has been commonly reported in the literature [32]. Some common co-catalysts include alumina and potassium oxide. MoP is commonly used as an efficient catalyst for hydrodesulfurization (HDS) and hydrodenitrogenation (HDN) reactions [33]. Depending on the reversibility of hydrogen bonding to the catalyst, some catalysts used for HDS reactions are also useful for HER reactions because of the similar pathways and mechanisms of hydrogen production and hydrogenation as well as their low Tafel slope and low over potential. For example, Chen et al. [34] impregnated precursors on sponges to obtain MoP with a large specific surface area and enhanced photocatalytic activity. MoP cannot be directly used as a photocatalyst, but it can be used as an efficient hydrogen precipitation co-catalyst. Du et al. used MoP as a highly active co-catalyst on CdS nanorods for the first time, which significantly improved the photocatalytic activity of their CdS catalyst [35]. Thus, MoP is an efficient co-catalyst for hydrogen precipitation.

At present, the utilization of solar energy by metal oxide photocatalysts for hydrogen production has mainly focused on the UV wavelength range. Furthermore, most research is based on TiO_2_ semiconductor photocatalytic materials. The majority of the wavelengths that make up solar energy, though, do not fall inside the visible spectrum. ZnIn_2_S_4_ shows promise as a visible-light-responsible ternary metal–sulfur compound photocatalyst, but its performance still needs to be improved. Therefore, in this work, ZnIn_2_S_4_ materials were prepared and modified (as shown in Figure 1): (1) Petal-shaped ZnIn_2_S_4_ catalysts were produced, their morphology was studied, and their photocatalytic performance was investigated. (2) Co-ZnIn_2_S_4_ was prepared by Co doping and ultrasonic exfoliation to broaden the absorption band edge and retain the petal-shaped morphology of the catalyst. (3) An a-TiO_2_/Co-ZnIn_2_S_4_ composite photocatalyst was successfully prepared by coating amorphous TiO_2_ on the Co-ZnIn_2_S_4_ surface, and the effect of loading different amounts of TiO_2_ on the photocatalytic performance was investigated. At the same time, a TiO_2_ and Co-ZnIn_2_S_4_ heterojunction was constructed, which led to the red-shift of the absorption band, enhanced light absorption properties, and a reduction in photogenerated electron–hole recombination. (4) Finally, MoP was loaded on the a-TiO_2_/Co-ZnIn_2_S_4_ catalyst as a co-catalyst, which enhanced the light absorption intensity and provided reaction sites to promote the overall efficiency of catalytic hydrogen production. Therefore, MoP/a-TiO_2_/Co-ZnIn_2_S_4_ flower-like composite photocatalysts with good photocatalytic hydrogen production activity and stability were prepared. This catalyst uses Co-ZnIn_2_S_4_ as the main body for photo generated electron excitation, and amorphous a-TiO_2_ is combined with it to improve the efficiency of electron hole separation. Finally, MoP is used as a co-catalyst to provide hydrogen production sites, thus achieving efficient hydrogen production.

## 2. Results and Discussions

### 2.1. Structure, Morphology and Composition of Composite Photocatalysts

The synthesized ZnIn_2_S_4_ and loaded catalyst samples were characterized to investigate their morphology and microstructures, as shown by the SEM images in Figure 2. Figure 2a shows that the synthesized ZnIn_2_S_4_ was a petal-like microsphere consisting of a large number of nanoflakes, which are all made of ZnIn_2_S_4_ nanosheets. These nanoflakes were cross-linked to each other and formed many uniform slit-type pore structures between the petal layers. Figure 2b shows that Co doping did not change the flower-like structure of ZnIn_2_S_4_. No particles of Co aggregation were observed on the surface of the petals, so this demonstrated that Co was potentially doped into the lattice structure of ZnIn_2_S_4_. Figure 2c shows that the petals were loaded with a granular material, which indicated the successful loading of TiO_2_. This was consistent with the catalyst morphology design. As shown in Figure 2d, the addition of MoP did not result in any obvious morphological changes. However, the MoP content was repeatedly low.

Element mapping (Figure 3 and Table 1) confirmed that MoP/TiO_2_/Co-ZnIn_2_S_4_ contained S, Mo, In, Zn, Ti, O, P, and Co elements. All elements were evenly distributed without visible aggregation, further demonstrating the successful synthesis of the MoP/a-TiO_2_/Co-ZnIn_2_S_4_ composite.

The morphological characteristics of the MoP/a-TiO_2_/Co-ZnIn_2_S_4_ photocatalyst were further investigated by TEM, as shown in Figure 4. Figure 4a shows that the MoP/a-TiO_2_/Co-ZnIn_2_S_4_ composite system had a nanoflower-like structure and intact, non-agglomerated microspheres. Figure 4b is a partial enlargement of Figure 4a, showing a more detailed view of the ZnIn_2_S_4_ nanosheets, which are very thin in the nanoflower. Some MoP/TiO_2_ particles were visible on the nanosheets, which showed the successful loading of MoP and TiO_2_ on the surface of ZnIn_2_S_4_. Figure 4c shows an electron diffraction pattern of the MoP/a-TiO_2_/Co-ZnIn_2_S_4_ photocatalyst, demonstrating its good crystallinity. Lattice fringe spacings of 0.21, 0.32 and 0.35 nm were identified in Figure 4d, which respectively corresponded to MoP, ZnIn_2_S_4_, and TiO_2_. This was consistent with the data in the relevant literature. Overall, this TEM analysis further demonstrated the successful preparation of MoP/a-TiO_2_/Co-ZnIn_2_S_4_.

The catalyst samples were investigated by X-ray diffraction, as shown in Figure 5. Characteristic diffraction peaks were visible at 8.52°, 21.3°, 29.1°, and 49.3° for all four catalysts, and these peaks were consistent with standard cards JCPDS 49–1562 and JCPDS 48–1778. ZnIn_2_S_4_ is a direct bandgap semiconductor with a layered (according to card NO. 48–1778, a = b = c = 10.6,α = β = γ = 90°) and trigonal structure (ICSD-JCPDS card NO. 49–1562, a = b = 3.85, c = 24.68, α = 37.01°,β = 90°,γ = 120°), as shown in Figure 5. All polymorphs show certain photocatalytic performance under visible light, while the hexagonal ZnIn_2_S_4_ has better photocatalytic performance. The cubic ZnIn_2_S_4_ is a direct cubicspinel phase when the S atoms in the unit cell are ABC stacking [36]. The diffraction peak positions did not significantly shift upon modification, which indicated that the ZnIn_2_S_4_ was not significantly affected. The shape of the ZnIn_2_S_4_ peaks did not change after TiO_2_ loading and no separate TiO_2_ peaks were identified, indicating that partly amorphous TiO_2_ was synthesized. The diffraction peaks also did not change after the addition of MoP, indicating the successful preparation of the composite MoP/a-TiO_2_/Co-ZnIn_2_S_4_ catalyst.

Figure 6a,b shows the N_2_ adsorption–desorption isotherms and pore size distributions of ZnIn_2_S_4_, Co-ZnIn_2_S_4_, a-TiO_2_/Co-ZnIn_2_S_4_ and MoP/a-TiO_2_/Co-ZnIn_2_S_4_. All four isotherms were identified as type IV, and they contained H3 hysteresis loops. Moreover, the catalysts exhibited pore sizes ranging from 10 to 100 nm. As shown in Table 2, the specific surface area slightly changed after catalyst modification. Specifically, MoP/a-TiO_2_/Co-ZnIn_2_S_4_ showed a slight increase in specific surface area compared with pure ZnIn_2_S_4_. The pore volume of MoP/a-TiO_2_/Co-ZnIn_2_S_4_ was also slightly higher than that of pure ZnIn_2_S_4_. This was consistent with the SEM image shown in Figure 2c, in which some of the TiO_2_ nanoparticles were supported on the ZnIn_2_S_4_ nanosheets. Therefore, the addition of TiO_2_ and the MoP co-catalyst led to an increase in adsorption pore volume. Consequently, more adsorption and active sites were generated on the photocatalyst surface. Moreover, the modified catalysts exhibited lower pore sizes because TiO_2_ was distributed between the ZnIn_2_S_4_ petals, which reduced the pore size.

The chemical state and chemical composition of the MoP/a-TiO_2_/Co-ZnIn_2_S_4_ composite was analyzed by XPS. As shown in Figure 7, the XPS survey spectrum confirmed the presence of P, Mo, Ti, Zn, In and S elements in this composite photocatalyst. This was consistent with the EDS test results. The binding energy peaks at 445.1 eV, 225.9 eV, 139.8 eV and 161.9 eV were attributed to In3d, Mo3d, P2p and S 2p signals, respectively. This indicated the presence of MoP, TiO_2_ and ZnIn_2_S_4_ in the MoP/a-TiO_2_/Co-Znln_2_S_4_ composite photocatalyst. In the Ti 2p spectrum, the two main peaks near 458.7 eV and 464.5 eV were attributed to Ti 2p_3/2_ and Ti 2p_1/2_.These peaks were generated by the Ti^4+^ oxidation state of TiO_2_. A single O1s peak near 530.0 eV was deconvoluted into three peaks. The peak at 530.0 eV was attributed to the presence of oxygen vacancies, and the peaks at 530.8 and 532.4 eV were caused by Ti–OH. This demonstrated that the synthesized TiO_2_ was amorphous and that the presence of this TiO_2_ increased the oxygen vacancy concentration of the catalyst. These results conclusively demonstrate that the MoP/a-TiO_2_/Co-ZnIn_2_S_4_ composite photocatalyst was successfully prepared.

UV-vis diffuse reflectance spectra of TiO_2_/Co-ZnIn_2_S_4_ were obtained using different TiO_2_ loading times to explore the effect of TiO_2_on catalytic activity, as shown in Figure 8a. These spectra were denoted as X-TiO_2_/Co-ZnIn_2_S_4_, where X represents the number of minutes. With increasing TiO_2_ loading time, the light absorption intensity and range of this composite catalyst first increased and then decreased. In particular, at 20 min, the absorption side band of a-TiO_2_/Co-ZnIn_2_S_4_ shifted to the right, and the highest absorption was achieved at this time. Therefore, 20-TiO_2_/Co-ZnIn_2_S_4_ was selected as the basis for subsequent experiments. Figure 8b shows UV-vis diffuse reflectance spectra of ZnIn_2_S_4_, Co-ZnIn_2_S_4_, a-TiO_2_/Co-ZnIn_2_S_4_ and MoP/a-TiO_2_/Co-ZnIn_2_S_4_. The absorption edge of pure ZnIn_2_S_4_ synthesized in this study was 480 nm. As shown, the spectrum significantly changed after Co doping, reaching 500 nm. However, almost the same absorption was demonstrated after the addition of amorphous TiO_2_. The absorption edge of MoP/a-TiO_2_/Co-ZnIn_2_S_4_ was widened to about 518 nm, indicating that the MoP was photocatalyzed by the load. The band gap energies of ZnIn_2_S_4_, Co-ZnIn_2_S_4_, a-TiO_2_/Co-ZnIn_2_S_4_ and MoP/a-TiO_2_/Co-ZnIn_2_S_4_ were calculated using the curves shown in Figure 8b. As shown in Figure 8c, the band gap energy of MoP/a-TiO_2_/Co-ZnIn_2_S_4_ was 2.7 eV. This analysis demonstrated the broader light absorption range and enhanced photocatalytic activity of the MoP/a-TiO_2_/Co-ZnIn_2_S_4_ composite photocatalyst.

### 2.2. Photoelectrochemical Performance

The photoelectrical properties of the catalysts were characterized in order to study their photocatalytic activity. Figure 9a shows that the photocurrent starting positions of all the catalysts were significantly earlier than their dark current starting positions. In addition, compared with Co-ZnIn_2_S_4_, the initial positions of TiO_2_/Co-ZnIn_2_S_4_ and MoP/a-TiO_2_/Co-ZnIn_2_S_4_ were slightly shifted to the left. These results indicate that the composite catalyst had a lower activation energy and enhanced photocatalytic performance. Linear scan voltammetry curves of ZnIn_2_S_4_, Co-ZnIn_2_S_4_, a-TiO_2_/Co-ZnIn_2_S_4_, and MoP/a-TiO_2_/Co-ZnIn_2_S_4_ were obtained under both light and dark conditions, as shown in Figure 9b. This shows the photocurrent densities of ZnIn_2_S_4_, Co-ZnIn_2_S_4_, a-TiO_2_/Co-ZnIn_2_S_4_ and MoP/a-TiO_2_/Co-ZnIn_2_S_4_, which exhibited photocurrents of 1 μA/cm^2^, 3 μA/cm^2^, 4 μA/cm^2^ and 4.5 μA/cm^2^, respectively. This showed that Co doping significantly improved the performance of Znln_2_S_4_. Moreover, the separation efficiency of photogenerated electron–hole pairs was also significantly improved by the addition of TiO_2_ supported on the ZnIn_2_S_4_ nanosheets. In addition, MoP/a-TiO_2_/Co-ZnIn_2_S_4_ had a higher photocurrent density, therefore exhibiting more efficient carrier separation and transfer efficiency. The electrochemical impedance spectra shown in Figure 9c demonstrate that MoP/a-TiO_2_/Co-ZnIn_2_S_4_ had lower high-frequency semicircles than Co-ZnIn_2_S_4_ or a-TiO_2_/Co-ZnIn_2_S_4_ as well as lower resistance. This further indicated that MoP/a-TiO_2_/Co-ZnIn_2_S_4_ had more efficient carrier separation and transfer efficiency. The line increase of curves Co-ZnIn_2_S_4_, a-TiO_2_/Co-ZnIn_2_S_4_ and MoP/a-TiO_2_/Co-ZnIn_2_S_4_ indicates that the charge transfer resistance decreases sequentially, which is also consistent with the higher carrier separation and transfer efficiency.

### 2.3. Photocatalytic Hydrogen Production Performance

The hydrogen production performance of the prepared catalysts was investigated, as shown in Figure 10. With increasing illumination time, hydrogen production increased for all four catalysts. As shown in Figure 10a, to explore the influence of the supported TiO_2_ on catalytic activity, the photocatalytic hydrogen production performances of a-TiO_2_/Co-ZnIn_2_S_4_ samples prepared by loading TiO_2_ for different amounts of time were also investigated. The optimal hydrogen production rate of 3.88 mmol·g^−1^ was achieved by using 20-TiO_2_/Co-ZnIn_2_S_4_, which was consistent with the UV-vis spectra shown in Figure 8a. The hydrogen production rates of ZnIn_2_S_4_, Co-ZnIn_2_S_4_, a-TiO_2_/Co-ZnIn_2_S_4_, MoP/a-TiO_2_/Co-ZnIn_2_S_4_ and P25 are shown in Figure 10b. As can be seen, Co doping, the addition of TiO_2_, and the addition of MoP all led to enhanced hydrogen evolution. It also can be seen from Figure 10b that the hydrogen production using non-noble metal co-catalyst MoP (7.42 mmol·g^−1^) is approximately twice as much as using Pt co-catalyst (3.88 mmol·g^−1^). In addition, it can be observed from Figure 10b,c that the hydrogen production capacity and hydrogen production rate of Pt/P25 at 2.5 h is 6.43 mmol/g and 2.55 mmol·h^−1^·g^−1^, respectively. As shown in Figure 10c, the highest hydrogen production rate of 2.96 mmol·h^−1^·g^−1^ was achieved by MoP/a-TiO_2_/Co-ZnIn_2_S_4_; in contrast, the sample Pt/a-TiO_2_/Co-ZnIn_2_S_4_ achieved 1.55 mmol·h^−1^·g^−1^. Each modification step (Co doping, addition of supported TiO_2_, addition of MoP co-catalyst) further enhanced the hydrogen evolution rate compared with the unmodified ZnIn_2_S_4_. These results showed that the MoP/a-TiO_2_/Co-ZnIn_2_S_4_composite photocatalyst had excellent hydrogen production performance. It can be seen that the hydrogen production efficiency of the MoP/a-TiO_2_/Co-ZnIn_2_S_4_ without Pt is still higher in comparison to that of P25 powder.

Photocatalytic stability across multiple cycles is another important factor that influences the practical application of photocatalysts. Therefore, the MoP/a-TiO_2_/Co-ZnIn_2_S_4_ composite catalyst was tested for cyclic hydrogen production. As shown in Figure 10d, after use in three cycles, the hydrogen production only decreased by 5%, indicating that it has good cycle stability. However, after 5 cycles, a slight decline in activity was observed (decline rate of 13.5%). This indicated a certain degree of catalytic stability. The degradation between cycles 3 and 4 (decline rate of 4.2%) was greater than that between cycles 1 and 2 (decline rate of 2.1%) due to the photocorrosion of ZnIn_2_S_4_.

### 2.4. Mechanism of Photocatalytic Hydrogen Evolution

According to the band gap structures and Fermi level of TiO_2_ and ZnIn_2_S_4_, the possible transfer processes of photogenerated electron–hole pairs are proposed in Figure 11 [37]. The photogenerated electrons in the CB of ZnIn_2_S_4_ migrate to the CB of TiO_2_ while the photoexcited holes in the VB of TiO_2_ transfer to the VB of ZnIn_2_S_4_. The E_CB_ of the photogenerated electrons is lower than the E_0_ redox (H^+^/H_2_). The presumed process is, therefore, not feasible in this photocatalytic process. Another possible reaction mechanism is shown in Figure 11. In the photocatalytic reaction, the solid–solid contact interface between ZnIn_2_S_4_ and TiO_2_ serves as the combination center of the photogenerated electrons in the CB of TiO_2_ and the photogenerated holes in the VB of ZnIn_2_S_4_ [38]. The photogenerated electrons involved in the reaction have a stronger reduction ability than that of pure TiO_2_, thus performing a better photocatalytic activity for HER. The photogenerated holes in the VB of TiO_2_ oxidize water to O_2_, while the photogenerated electrons in the CB of ZnIn_2_S_4_ simultaneously reduce H^+^ to H_2_. In summary, all of the above analyses show that the electron transfer process is identified as an S-scheme mechanism in this study.

## 3. Experimental Section

### 3.1. Materials and Characterization

Zinc chloride (ZnCl_2_, AR), indium chloride (InCl_3_, AR), thioacetamide (TAA, AR), nickel chloride hexahydrate (NiCl_2_·6H_2_O, AR), tungsten chloride (WCl_6_, AR), polyethylene glycol (HO(CH_2_CH_2_O)_n_H, AR), melamine (C_3_N_3_(NH_2_)_3_, AR), and sodium citrate dehydrate (C_6_H_5_Na_3_O_7_·2H_2_O, AR) were purchased from Shanghai Macklin Biochemical Co. (Shanghai, China) Triethanolamine (TEOA, AR) was purchased from Tianjin Beichen Founder Reagent Factory (Tianjin, China). Ethyl alcohol (CH_3_CH_2_OH, AR) was provided by Sinopharm Chemical Reagent Co., Ltd. (Shanghai, China). All materials were used as received.

Sample morphologies were analyzed using scanning electron microscopy (SEM, JSM-7900F, JEOL, Tokyo, Japan) coupled with energy-dispersive X-ray spectroscopy (OXFORD MAX-80, Oxford, UK). Transmission electron microscopy (TEM) was performed using a JSM-2100plus (JEOL, Tokyo, Japan). X-ray diffraction (XRD, Bruker (Billerica, MA, USA), D8 Advance) was used for crystal structure analysis. XRD patterns were obtained in the 2θ range of 20–90° with a scanning rate of 6°/min. Surface compositions were investigated by X-ray photoelectron spectroscopy (XPS) using an AMICUS ESCA3200 (Philadelphia, PA, USA). The XPS spectra were corrected using the C1s peak at 284.8 eV. Ultraviolet-visible (UV-vis) diffuse reflectance spectra (DRS) were obtained in the 200–800 nm range by a UV-vis spectrophotometer (Shimadzu UV-2450, Kyoto, Japan). Photoluminescence (PL) spectra were collected using an Perkin-Elmer LS50B (Buckinghamshire, UK) with a 380 nm excitation wavelength at room temperature. BET surface areas and porosity were measured via nitrogen adsorption–desorption experiments using a Micromeritics ASAP 2020 (Micromeritics, Norcross, GA, USA).

### 3.2. Steps for Preparation of MoP/a-TiO_2_/Co-ZnIn_2_S_4_ Flower-like Composite Photocatalysts

#### 3.2.1. Preparation of Co-ZnIn_2_S_4_ Catalyst

A total of 0.136 g zinc chloride, 0.586 g indium chloride, and 0.301 g thioacetamide were weighed and added to 80 mL ethylene glycol. This mixture was stirred and centrifugally sonicated to dissolve the solid compounds. The solution was then transferred to a 100 mL hydrothermal kettle and heated in an oven at 180 °C for 2 h. After the reaction, the reaction solution was removed from the hydrothermal kettle and left to stand for 0.5 h. Next, centrifugation was used to obtain the solid product. The sample was then crushed with agate mortar to obtain ZnIn_2_S_4_. Doped Co-ZnIn_2_S_4_ was obtained by repeating this experimental procedure with the addition of 0.0069 g Co(NO_3_)_2_·6(H_2_O).

#### 3.2.2. Preparation of a-TiO_2_/Co-ZnIn_2_S_4_ Catalyst

A total of 80 mg Co-ZnIn_2_S_4_ was added to 20 mL isopropanol. Then, 100 μL tetrabutyltitanate and 20 μL water were then added dropwise under stirring. Five samples were prepared and stirred for 5 min, 10 min, 20 min, 30 min and 60 min for the control test. These samples were centrifuged three times using isopropanol and then dried at 60 °C in an oven. The dried samples were ground and heated in a muffle furnace at 120 °C for 1 h to obtain a-TiO_2_/Co-ZnIn_2_S_4_.

#### 3.2.3. Preparation of MoP/a-TiO_2_/Co-ZnIn_2_S_4_ Catalyst

A mixture of 1 g Na_2_MoO_4_·2H_2_O and 10 g NaH_2_PO_2_ was ground in a mortar for 0.5 h until no crystal particles remained, and then transferred to a tubular furnace, under Ar protection at 400 °C (heating rate 10 °C/min), and calcined for 1 h to obtain MoP. A 2 mg/mL MoP solution was then prepared. Next, 0.5 g a-TiO_2_/Co-ZnIn_2_S_4_ and 1.25 mL MoP solution were magnetically stirred for 0.5 h. MoP/a-TiO_2_/Co-ZnIn_2_S_4_ was obtained by drying the resulting product in an oven at 60 °C followed by crushing with a mortar.

### 3.3. Photocatalysis and Photoelectrochemical Performance Measurements

#### 3.3.1. Hydrogen Production Performance

Photocatalytic experiments were performed using an online photocatalytic hydrogen evolution system (Meiruichen, Beijing, China MC-SCO_2_II-AG) at 5 °C using a 300 W Xe lamp equipped with a AM1.5G cutoff filter positioned 20 cm away from the reactor. A total of 10 mg of catalyst was dispersed in 100 mL of 0.1 M Na_2_S and 0.1 M Na_2_SO_3_ solution and the mixture was stirred in vacuum for 30 min. We first ran tests in the dark for one hour to confirm no H_2_ production. Hydrogen evolution, detected by an online gas chromatography (using FL9790, Fuli, Zhjiang, TCD with nitrogen as a carrier gas and 5 Å molecular sieve column) was observed only under light irradiation. At the end of the photocatalytic reaction, which lasted for 2.5 h, the reactor was refilled with 10 mL of Na_2_S and Na_2_SO_3_ solutions and degassed. Then, 10 mg of P25 was dispersed in 100 mL of CH_3_OH/H_2_O solution, and 0.1 mL (1 mg/mL) of chloroplatinic acid was added.

#### 3.3.2. Photoelectrochemical Performance

A CHI760E electrochemical workstation and a standard three-electrode system (platinum sheet, saturated silver chloride electrode, and the loaded FTO substrate) were used to analyze the catalysts. An aqueous 0.5 mol/L Na_2_SO_4_ solution was used as the electrolyte for transient photocurrent testing and electrochemical impedance spectroscopy (EIS) testing. A xenon lamp light source system with an AM1.5G filter was used to simulate daylight for photocurrent testing.

Photocatalytic hydrogen production experiments were performed using a vacuum photocatalytic carbon dioxide reduction system with a xenon light source to simulate daylight. A working electrode for photochemical measurements was prepared using 7.5 mg of sample (ZnIn_2_S_4_, Co-ZnIn_2_S_4_, a-TiO_2_/Co-ZnIn_2_S_4_ and MoP/a-TiO_2_/Co-ZnIn_2_S_4_), which was sonicated for 30 min in a mixture containing 375 μL of ultrapure water, 125 of ethanol and 30 μL of naphthol. Then, 30 μL of the resulting suspension was used to drop-coat FTO glass, which was then heated for 30 min at 300 °C under Ar. A 1 cm × 1 cm glass substrate was then ultrasonicated first in acetone, then in ethanol and finally in water (15 min each step), after which it was dried by a flow of Ar. PEC measurements were conducted using a single compartment quartz cell with three electrodes. Data were recorded by the workstation equipment containing photoanode, saturated Ag/AgCl and 1 cm × 1 cm Pt piece as working, reference and counter electrodes, respectively. We used 0.2 M Na_2_SO_4_ with pH = 6.5 as electrolyte. PEC tests were conducted using a 150 W Xenon lamp equipped with a standard AM 1.5G filter. Quartz cell was positioned 10 cm away from the light source. The recorded potential was converted to reference hydrogen electrode (RHE) potentials using the following equation: $$E_{\mathrm{RHE}}=E_{\mathrm{Ag}/\mathrm{AgCl}}+\mathrm{pH}\;*\;0.059+0.195\;V$$

Linear sweep voltammetry (LSV) was conducted at 10 mV/s scan rate in the −0.4–1.2 V scan range relative to the Ag/AgCl electrode. EIS was performed under Xe lamp at 0 V with AC potential ranging from 100 K to 0.1 Hz.

## 4. Summary

In summary, a MoP/a-TiO_2_/Co-ZnIn_2_S_4_ composite photocatalyst was successfully prepared by a facile hydrothermal method. The Co dopant in the flower-like ZnIn_2_S_4_ broadened the absorption band edge of the composite catalyst. Amorphous TiO_2_ and Co-ZnIn_2_S_4_ were combined to form a heterojunction, which improved the photocarrier separation efficiency and the stability of the catalyst. More importantly, the introduction of amorphous TiO_2_ created oxygen vacancies, which further improved the carrier density. Finally, the non-noble metal catalyst MoP nanoparticles were introduced into the system as co-catalysts, which became the hydrogen production sites and realized high-efficiency hydrogen production. The flower-like MoP/a-TiO_2_/Co-ZnIn_2_S_4_ composite photocatalyst exhibited a hydrogen production rate of 2.96 mmol·h^−1^·g^−1^, which was 0.98 mmol·h^−1^·g^−1^ of that of the pure ZnIn_2_S_4_. Therefore, this catalyst shows great promise for the green production of hydrogen. Moreover, this study also provides new insight into the design and underlying mechanism of direct Z-schemes for enhanced photocatalysis.

## Figures and Tables

**Figure 1 molecules-28-04350-f001:**
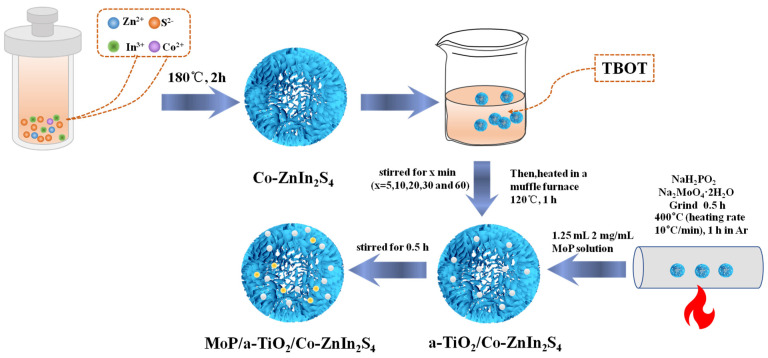
Steps for preparation of MoP/a-TiO_2_/Co-ZnIn_2_S_4_ flower-like composite photocatalysts.

**Figure 2 molecules-28-04350-f002:**
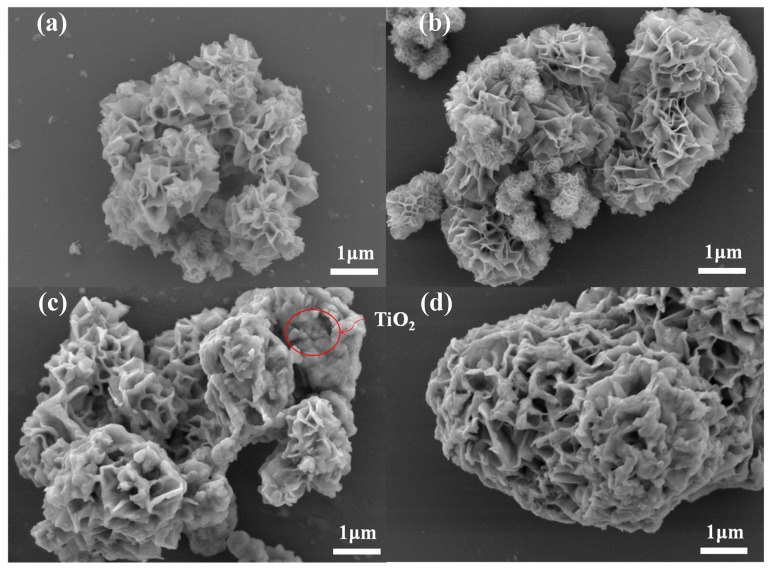
SEM micrographs of (**a**) ZnIn_2_S_4_, (**b**) Co-ZnIn_2_S_4_, (**c**) a-TiO_2_/Co-ZnIn_2_S_4_, and (**d**) MoP/a-TiO_2_/Co-ZnIn_2_S_4_.

**Figure 3 molecules-28-04350-f003:**
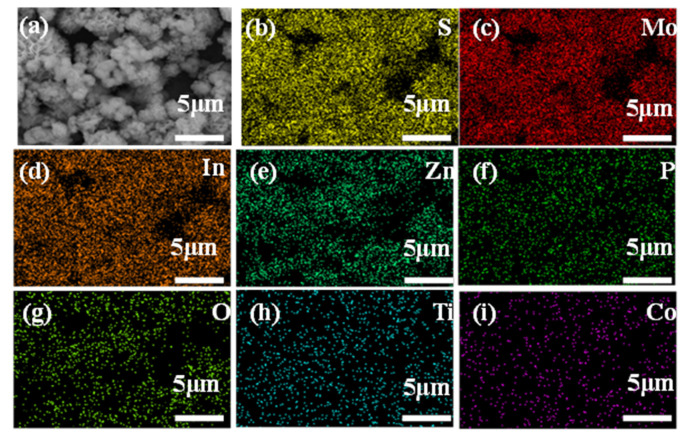
Elemental mapping of the MoP/a-TiO_2_/Co-ZnIn_2_S_4_ composite catalyst. (**a**) scanning area, (**b**) S, (**c**) Mo, (**d**) In, (**e**) Zn, (**f**) P, (**g**) O, (**h**) Ti, (**i**) Co.

**Figure 4 molecules-28-04350-f004:**
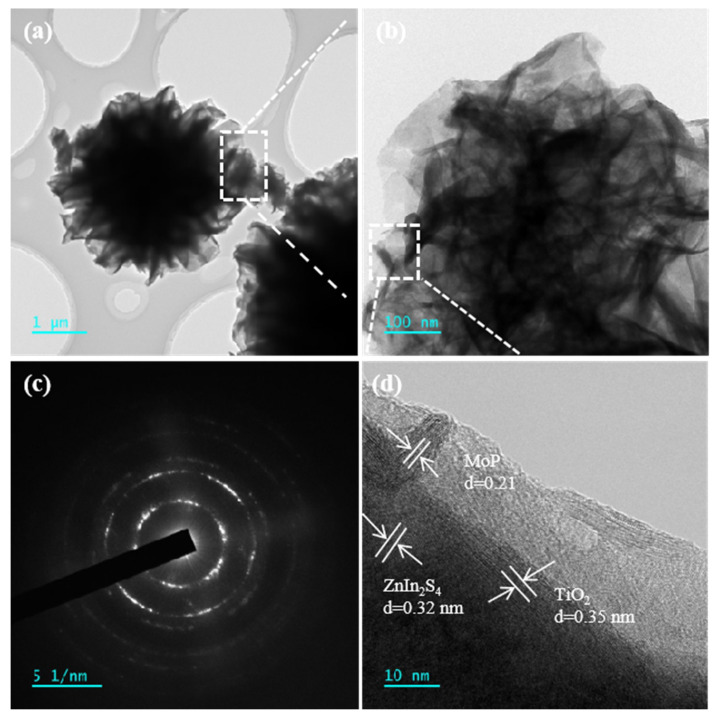
(**a**,**b**) TEM micrographs, (**c**) electron diffraction pattern, and (**d**) high-resolution TEM micrograph showing the lattice fringe spacing of MoP/a-TiO_2_/Co-ZnIn_2_S_4_.

**Figure 5 molecules-28-04350-f005:**
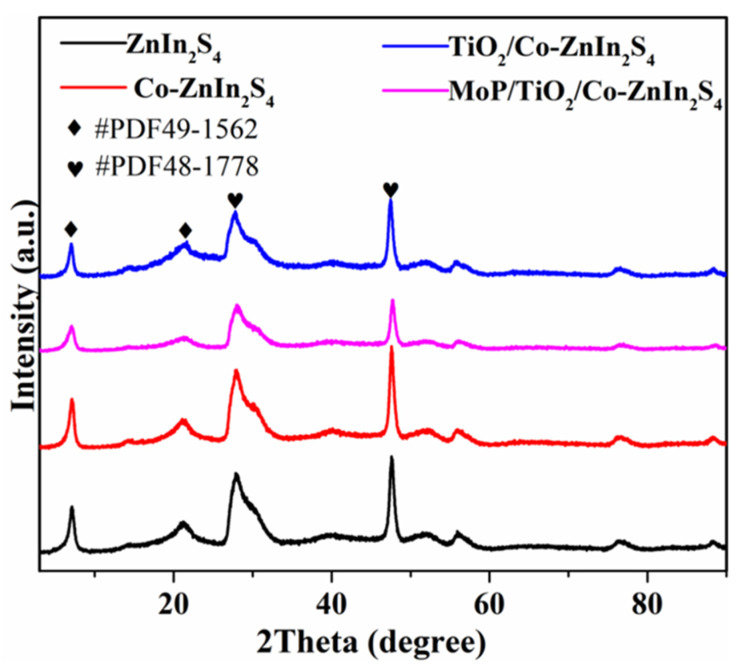
XRD patterns of the ZnIn_2_S_4_, Co-ZnIn_2_S_4_, a-TiO_2_/Co-ZnIn_2_S_4_, and MoP/a-TiO_2_/Co-ZnIn_2_S_4_, photocatalysts.

**Figure 6 molecules-28-04350-f006:**
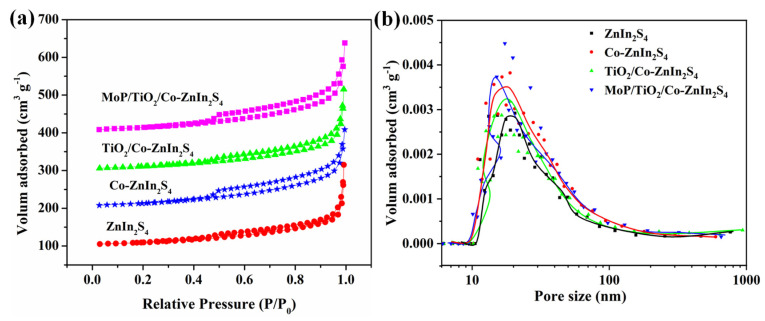
(**a**) N_2_ adsorption–desorption isotherms and (**b**) pore size distributions of the ZnIn_2_S_4_, Co-ZnIn_2_S_4_, a-TiO_2_/Co-ZnIn_2_S_4_, and MoP/a-TiO_2_/Co-ZnIn_2_S_4_ photocatalysts.

**Figure 7 molecules-28-04350-f007:**
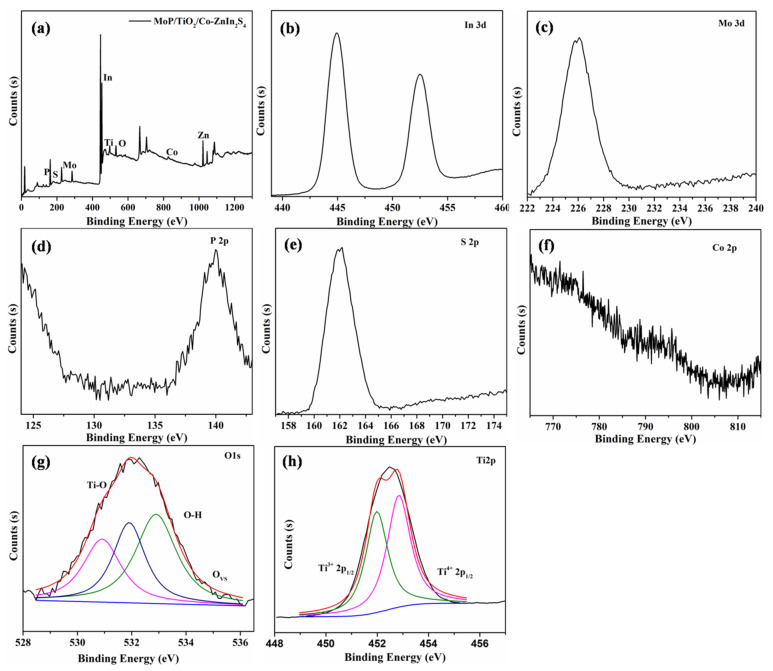
XPS spectra of the MoP/a-TiO_2_/Co-ZnIn_2_S_4_ composite catalyst.

**Figure 8 molecules-28-04350-f008:**
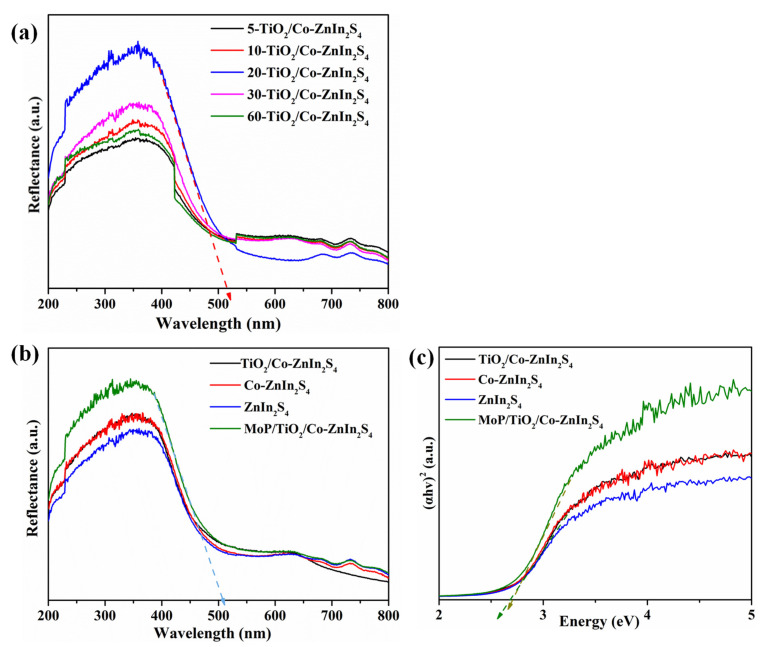
(**a**) UV-vis diffuse reflectance spectra of a-TiO_2_/Co-ZnIn_2_S_4_ at different times; (**b**) UV-vis diffuse reflectance spectra and (**c**) bandgaps of ZnIn_2_S_4_, Co-ZnIn_2_S_4_, a-TiO_2_/Co-ZnIn_2_S_4_, and MoP/a-TiO_2_/Co-ZnIn_2_S_4_.

**Figure 9 molecules-28-04350-f009:**
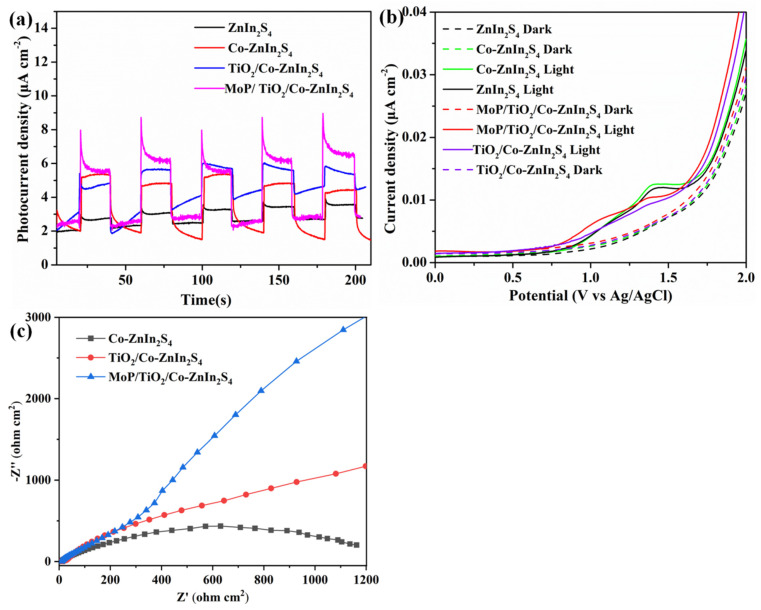
(**a**) Photocurrent response curves of Co-ZnIn_2_S_4_, a-TiO_2_/Co-ZnIn_2_S_4_, and MoP/a-TiO_2_/Co-ZnIn_2_S_4_; (**b**) linear scanning voltammograms and photocurrent response curves of ZnIn_2_S_4_, Co-ZnIn_2_S_4_, a-TiO_2_/Co-ZnIn_2_S_4_, and MoP/a-TiO_2_/Co-ZnIn_2_S_4_; (**c**) EIS curves of Co-ZnIn_2_S_4_, a-TiO_2_/Co-ZnIn_2_S_4_, and MoP/a-TiO_2_/Co-ZnIn_2_S_4_.

**Figure 10 molecules-28-04350-f010:**
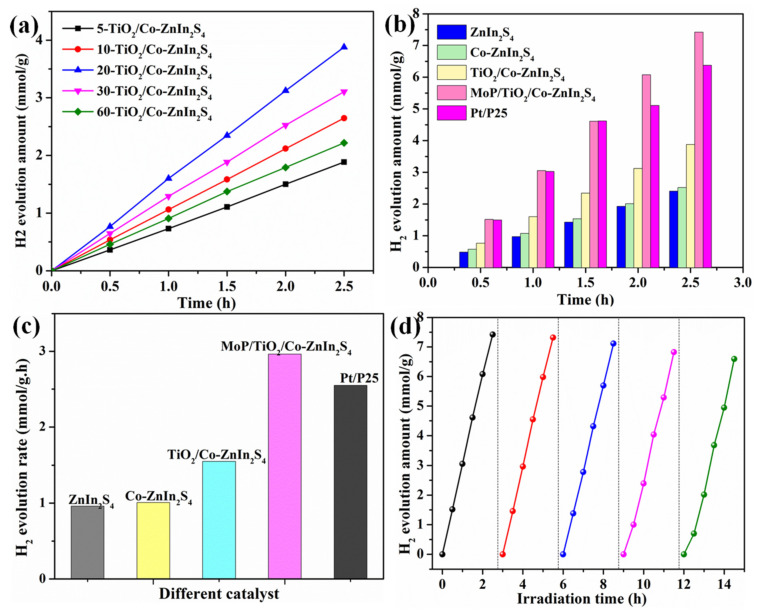
(**a**) Hydrogen production yields of a-TiO_2_/Co-ZnIn_2_S_4_ samples prepared in different amounts of time for TiO_2_ loading. (**b**) Hydrogen production yields of the prepared catalysts with increasing reaction time. (**c**) Hydrogen production rates of the prepared catalysts. (**d**) Hydrogen production of MoP/a-TiO_2_/Co-ZnIn_2_S_4_ across 5 cycles.

**Figure 11 molecules-28-04350-f011:**
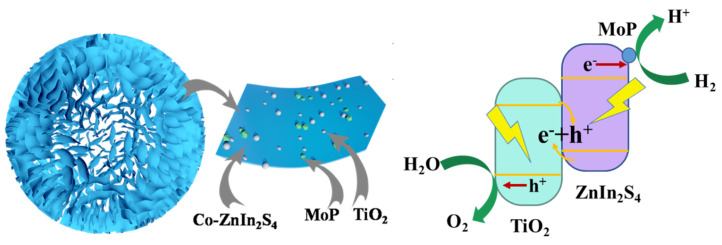
Composition diagram and photocatalytic reaction mechanism of MoP/a-TiO_2_/Co-ZnIn_2_S_4_.

**Table 1 molecules-28-04350-t001:** Distribution of elements in the MoP/a-TiO_2_/Co-ZnIn_2_S_4_ composite catalyst.

	In	S	Zn	Mo	P	Ti	O	Co
wt%	53.2	26.0	10.2	2.5	0.3	1.1	6.2	0.5

**Table 2 molecules-28-04350-t002:** Specific surface area, pore volume, and average pore diameter of ZnIn_2_S_4_, Co-ZnIn_2_S_4_, a-TiO_2_/Co-ZnIn_2_S_4_, and MoP/a-TiO_2_/Co-ZnIn_2_S_4_.

Sample	Specific Surface Area (m^2^/g)	Pore Volume (cm^3^/g)	Average Pore Diameter (nm)
ZnIn_2_S_4_	41.293	0.3346	162.1
Co-ZnIn_2_S_4_	53.453	0.3242	121.3
TiO_2_/Co-ZnIn_2_S_4_	46.669	0.3346	143.4
MoP/a-TiO_2_/Co-ZnIn_2_S_4_	53.250	0.3703	139.1

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
