# Peer review of "Synthesis and Hydrogen Production Performance of MoP/a-TiO2/Co-ZnIn2S4 Flower-like Composite Photocatalysts"

_molecules, 2023, doi:10.3390/molecules28114350_

Round 1

Reviewer 1 Report

1-     Lines 39 and 40. Please detail briefly the challenges related to hydrogen storage and transport.

2-     Line 44: how can hydrogen be produced using solar photolysis? If the authors are using photocatalysts, photocatalysis characteristics should be described.

3-     Line 57:  What does bandage mean in this context?

4-     Line 75: the term “longer wavelengths” is not appropriate. What does the authors mean? What is the wavelength range?

5-     The authors should mention if the combination of TiO2 and Co-ZnIn2S4-based photocatalysts been reported? The same applies for MoP/a-TiO2/Co-ZnIn2S4.

6-     Lines 113/114: the authors described the use of MoP for hydrogen precipitation. Has it been described before for photocatalytic hydrogen production?

7-     Line 130. What do the authors mean by electron-hole complexation?

8-     The authors mentioned amorphous TiO2 was incorporated to the material. Is there any study of amorphous TiO2 for hydrogen generation? It would be relevant to be included in the manuscript.

9-     The introduction section should contain information if MoP/a-TiO2/Co-ZnIn2S4.

10-  Line 203. The use of TiO2/WO3 was described as photoanode for photoelectrochemical tests. Why have ZnIn2S4, Co-ZnIn2S4, a-TiO2/Co-ZnIn2S4 and MoP/a-TiO2/Co-ZnIn2S4 materials been not compared?

11-  Line 184: what is the irradiance of the lamp?

12-  187: Was the electrolyte solution degassed for the experiments in dark conditions?

13-  Lines 225/226: Co was not identified in the SEM analysis. How can the authors infer Co was present in lattice structure using SEM?

14-  Figure 2C. There is no significant difference between Fig 2 a, b and c. What is the mean size of the flower-like structures in the material? How could “the successful loading” be described? TiO2 loading should be highlighted in the images. Figure 2C seems to be recorded at higher magnification than the other ones. Has the analysis been performed at the same magnification for all samples? Perhaps higher resolution and magnification micrographs would be suitable.

15-  The elemental mapping shows a much higher proportion of Co than the target of 0.3% mentioned in the introduction section. How do the authors explain it?

16-  Line 271: XRD analysis was not able to detect TiO2. The authors attributed it only to the low proportion in the material. Have the authors considered the TiO2 particle size?

17-  Figure 6B – pore size distribution. The plot should be done considering the trend of data collected instead of simply connecting the dots.

18-  Figure 8A – what does different times stand for? Would it be different times for TiO2 deposition? The legend should be updated. Additionally, the same colour code should be used for Figure 8B and 8C.

19-  Item 3.2 is described as photoelectric performance. The correct expression is photoelectrochemical performance.

20-  Fig 9A inset. The inset should show a smaller scale, so the semi-circle is more visible in the Nyquist plot. Why has the charge transfer resistance (RCT) not been discussed?

21-  How was Pt/P25 prepared for comparison? Details should be described in experimental section.

22-  Line 364. The hydrogen production rate was superior for 20-TiO2/Co-ZnIn2S4 than 30-TiO2/Co-ZnIn2S4. This is result is not consistent with Figure 8a UV-vis spectra. What is the reason for the change in behaviour?

23-   In Fig 10b discussion, “It also can be seen from figure 10b that the hydrogen production using non noble metal cocatalyst MoP (7.42 mmol·g-1) is twice more than that by using Pt cocatalyst (3.88 mmol·g-1)”. The data does not show twice more or doubled rate for any of the times studied.

24-  In Fig 10c discussion, the same material (MoP/a-TiO2/Co-ZnIn2S4) is being compared. Comparison with Pt/P25 should also be included.

25-  Fig 10d, stability should be discussed in terms of percentage drop or give a better understanding of the materials performance.

26-  Line 392. What is the reference used for amorphous Fermi level of TiO2? Please include a reference for this information.

Moderate editing of the English language is recommended

Author Response

Many thanks for your email and for giving us the revision opportunity on our manuscript entitled “Synthesis and hydrogen production performance of MoP/a-TiO2/Co-ZnIn2S4 flower-like composite photocatalysts” (Manuscript ID: molecules-2353011). Those comments are all valuable and very helpful for revising and improving our paper, as well as the important guiding significance to our researches. We tried our best to improve the manuscript and made some changes in the revised manuscript. We have listed changes at the end of this letter. And we would like to submit our corrected manuscript to “molecules”.

Comments and Suggestions for Authors

Reviewer1

  • Lines 39 and 40. Please detail briefly the challenges related to hydrogen storage and transport.

Reply: Thank you very much for your comment. We have been added in the manuscript and marked red.

Hydrogen's shortcomings are partly explained by high infrastructure costs for production, storage, and distribution. These problems may result from their low energy density per volume, explosive characteristics, and ability to cause embrittlement in metals such as steel[34].

  • Line 44: how can hydrogen be produced using solar photolysis? If the authors are using photocatalysts, photocatalysis characteristics should be described.

Reply: Thank you very much for your comment. We have been added in the manuscript and marked red.

Photogenerated charge carriers can be excited from photocatalysts under sunlight, and after the photogenerated electrons migrate to the surface of semiconductors, H+ in water receive electrons that are reduced to H2. The holes left behind are combined with sacrificial agents in the system and used to achieve continuous H2 production.

  • Line 57: What does bandage mean in this context?

Reply: Thank you very much for your comment. We have been checked in the manuscript and marked red. Please forgive our negligence.

Furthermore, the band gap of ZnIn2S4 is 2.3~2.5 eV, and the energy band of ZnIn2S4 is narrow, which is also conducive to the generation of photogenerated carriers[11].

  • Line 75: the term “longer wavelengths” is not appropriate. What does the authors mean? What is the wavelength range?

Reply: Thank you very much for your comment. We have been checked in the manuscript and marked red.

With increasing Co concentration, the absorption edge of the samples caused red shift, but the Co also gradually disrupted the ZnIn2S4 morphology.

The doping of Co ions can shift the absorption edge from 480 nm to 520 nm, which means moving towards the long wavelength direction.

  • The authors should mention if the combination of TiO2 and Co-ZnIn2S4-based photocatalysts been reported? The same applies for MoP/a-TiO2/Co-ZnIn2S4.

Reply: Thank you very much for your comment. We have researched the literature and found reports on Ni and Co double ion doping[1]. We have also found reports of TiO2 and ZnIn2S4 composite catalysts[2], but there have been no reports of using amorphous TiO2 and Co-ZnIn2S4 composite catalysts in this study. The use of MoP as a co catalyst in this system a-TiO2/Co-ZnIn2S4 has not been reported.

[1] Zhou D ,  Xue X ,  Wang X , et al. Ni, In co-doped ZnIn2S4 for efficient hydrogen evolution: Modulating charge flow and balancing H adsorption/desorption[J]. Applied Catalysis B: Environmental, 2022, 310:121337.

[2] Chen S ,  Yu J ,  Li S , et al. Study on the preparation of TiO2 nanofibers and the composition of ZnIn2S4/TiO2.  2017.

6-     Lines 113/114: the authors described the use of MoP for hydrogen precipitation. Has it been described before for photocatalytic hydrogen production?

Reply: Thank you very much for your comment. There have been many reports on the use of MoP as a co catalyst for photocatalytic hydrogen production[1]. Cheg et al synthetized nanosized MoP in a novel phosphorization process at a relatively low temperature under an ambient-air atmosphere and then assembled with g-C3N4 to form MoP/g-C3N4 coupled photocatalyst by a mixing and heat-treating method. The optimal MoP/g-C3N4 photocatalyst possessed a H2 production activity of 3.87 mmolh/g. But the use of MoP as a co catalyst in this system a-TiO2/Co-ZnIn2S4 has not been reported.

[1] Cheng C , Zong S , Shi J , et al. Facile preparation of nanosized MoP as cocatalyst coupled with g-C3N4 by surface bonding state for enhanced photocatalytic hydrogen production[J]. Applied Catalysis B: Environmental, 2020, 265:118620-.

7-     Line 130. What do the authors mean by electron-hole complexation?

Reply: Thank you very much for your comment. We have been checked in the manuscript and marked red. Please forgive our negligence.

It should be “recombination”, not “complexation”.

and a reduction in photogenerated electron-hole recombination.

8-     The authors mentioned amorphous TiO2 was incorporated to the material. Is there any study of amorphous TiO2 for hydrogen generation? It would be relevant to be included in the manuscript.

Reply: Thank you very much for your comment. We have been added in the manuscript and marked red.

Zywitzki D reported amorphous titania-based photocatalysts were synthesized using a facile, UV-light mediated method and evaluated as photocatalysts for hydrogen evolution from water/methanol mixtures. The resulting amorphous materials exhibit an overall higher hydrogen evolution rate (1.09 mmol·h-1·g-1 )compared to a crystalline TiO2 reference (P25 0.80 mmol·h-1·g-1) on a molar basis of the photocatalyst due to their highly porous structure and high surface area[36].

[36] D. Zywitzki, H. Jing, H. Tüysüz, et al. High surface area, amorphous titania with reactive Ti3+ through a photo-assisted synthesis method for photocatalytic H2 generation[J]. J. Mater. Chem. A, 2017(22): 10957 - 10967.

9-     The introduction section should contain information of MoP/a-TiO2/Co-ZnIn2S4.

Reply: Thank you very much for your comment. This study reports for the first time a composite catalyst MoP/a-TiO2/Co ZnIn2S4. The scientific ideas for its design and the technical route for its synthesis are detailed in the last paragraph of the introduction. At the same time, we have made relevant supplements according to your suggestion and marked red.

At present, the utilization of solar energy by metal oxide photocatalyst for hydrogen production has mainly focused on the UV wavelength range. Furthermore, most research is based on TiO2 semiconductor photocatalytic materials. The majority of the wavelengths that make up solar energy, though, do not fall inside the visible spectrum. ZnIn2S4 shows promise as a visible-light-responsible ternary metal-sulfur compound photocatalyst, but its performance still needs to be improved. Therefore, in this work, ZnIn2S4 materials were prepared and modified, as shown in Figure 1: (1) Petal-shaped ZnIn2S4 catalysts were produced, their morphology was studied, and their photocatalytic performance was investigated. (2) Co-ZnIn2S4 was prepared by Co doping and ultrasonic exfoliation to broaden the absorption band edge and retain the petal-shaped morphology of the catalyst. (3) An a-TiO2/Co-ZnIn2S4 composite photocatalyst was successfully prepared by coating amorphous TiO2 on the Co-ZnIn2S4 surface, and the effect of loading different amounts of TiO2 on the photocatalytic performance was investigated. At the same time, a TiO2 and Co-ZnIn2S4 heterojunction was constructed, which led to the red-shift of the absorption band, enhanced light absorption properties, and a reduction in photogenerated electron-hole recombination. (4) Finally, MoP was loaded on the a-TiO2/Co-ZnIn2S4 catalyst as a co-catalyst, which enhanced the light absorption intensity and provided reaction sites to promote the overall efficiency of catalytic hydrogen production. Therefore, MoP/a-TiO2/Co-ZnIn2S4 flower-like composite photocatalysts with good photocatalytic hydrogen production activity and stability were prepared. This catalyst uses Co-ZnIn2S4 as the main body for photo generated electron excitation, and amorphous a-TiO2 is combined with it to improve the efficiency of electron hole separation. Finally, MoP is used as a co catalyst to provide hydrogen production sites, thus achieving efficient hydrogen production.

10-  Line 203. The use of TiO2/WO3 was described as photoanode for photoelectrochemical tests. Why have ZnIn2S4, Co-ZnIn2S4, a-TiO2/Co-ZnIn2S4 and MoP/a-TiO2/Co-ZnIn2S4 materials been not compared?

Reply: Thank you very much for your comment. We have been checked in the manuscript and marked red. Please forgive our negligence.

Working electrode for photochemical measurements was prepared using 7.5 mg of sample (ZnIn2S4, Co-ZnIn2S4, a-TiO2/Co-ZnIn2S4 and MoP/a-TiO2/Co-ZnIn2S4),

11-  Line 184: what is the irradiance of the lamp?

Reply: Thank you very much for your comment. The radiance of xenon lamp measured by optical power meter under our experimental conditions (15cm, 200-800 nm) is 550w/m2.

12-  187: Was the electrolyte solution degassed for the experiments in dark conditions?

Reply: Thank you very much for your comment. Yes it was. Before the experiment, we degassed with ultrasound for five minutes and then cooled to room temperature.

13-  Lines 225/226: Co was not identified in the SEM analysis. How can the authors infer Co was present in lattice structure using SEM?

Reply: Thank you very much for your comment. We have been checked in the manuscript and marked red.

Figure 2b shows that Co doping did not change the flower-like structure of ZnIn2S4. No particles of Co aggregation were observed on the surface of the petals, so this demonstrated that Co maybe was doped into the lattice structure of ZnIn2S4.

14-  Figure 2C. There is no significant difference between Fig 2 a, b and c. What is the mean size of the flower-like structures in the material? How could “the successful loading” be described? TiO2 loading should be highlighted in the images. Figure 2C seems to be recorded at higher magnification than the other ones. Has the analysis been performed at the same magnification for all samples? Perhaps higher resolution and magnification micrographs would be suitable.

Reply: Thank you very much for your comment. We took new SEM images, and we have been replaced in the manuscript.

The mean size of the flower-like structures in the material is 5 μm. As shown in figure 2c, small nanoparticles between the petals can be seen, which indicates that TiO2 nanoparticles have been successfully loaded.

15-  The elemental mapping shows a much higher proportion of Co than the target of 0.3% mentioned in the introduction section. How do the authors explain it?

Reply: Thank you very much for your comment. We have been checked in the manuscript and marked red. Please forgive our negligence. We mistakenly wrote the Co content value in the EDS measured data when drawing the table.

Table1.Distribution of elements in the MoP/a-TiO2/Co-ZnIn2S4 composite catalyst

In

S

Zn

Mo

P

Ti

O

Co

Wt%

53.2

26.0

10.2

2.5

0.3

1.1

6.2

0.5

16-  Line 271: XRD analysis was not able to detect TiO2. The authors attributed it only to the low proportion in the material. Have the authors considered the TiO2 particle size?

Reply: Thank you very much for your comment. We have considered the possibility that amorphous TiO2 has a low content and small particle size, which exists in the form of nanodots in the sample, and has not been calcined to crystallize and loaded as amorphous. So no peak pattern of TiO2 was detected in XRD.

17-  Figure 6B – pore size distribution. The plot should be done considering the trend of data collected instead of simply connecting the dots.

Reply: Thank you very much for your comment. Following your suggestion, We manually fitted and plotted complex curves based on the distribution trend of the data. We have been replaced in the manuscript.

18-  Figure 8A – what does different times stand for? Would it be different times for TiO2 deposition? The legend should be updated. Additionally, the same colour code should be used for Figure 8B and 8C.

Reply: Thank you very much for your comment. We have been replaced in the manuscript.

Different times indicate the hydrolysis time of TBOT as a precursor, which can load different amounts of TiO2.

19-  Item 3.2 is described as photoelectric performance. The correct expression is photoelectrochemical performance.

Reply: Thank you very much for your comment. We have been checked in the manuscript and marked red.

Photoelectrochemical performanc

20-  Fig 9A inset. The inset should show a smaller scale, so the semi-circle is more visible in the Nyquist plot. Why has the charge transfer resistance (RCT) not been discussed?

Reply: Thank you very much for your comment. Please forgive our negligence. We retested the EIS and replaced the spectrogram.

The line increase of curves Co-ZnIn2S4, a-TiO2/Co-ZnIn2S4, and MoP/a-TiO2/Co-ZnIn2S4 indicates that the charge transfer resistance decreases sequentially, which is also consistent with the higher carrier separation and transfer efficiency.

21-  How was Pt/P25 prepared for comparison? Details should be described in experimental section.

Reply: Thank you very much for your comment. We have been added in the manuscript and marked red.

10 mg of P25 was dispersed in 100 mL of CH3OH/H2O solution, and 0.1mL (1mg/mL) of chloroplatinic acid was added.

22-  Line 364. The hydrogen production rate was superior for 20-TiO2/Co-ZnIn2S4 than 30-TiO2/Co-ZnIn2S4. This is result is not consistent with Figure 8a UV-vis spectra. What is the reason for the change in behaviour?

Reply: Thank you very much for your comment. In Figure 8a, the absorption edge value of 20-TiO2/Co ZnIn2S4 is 518nm, while the absorption edge value of 20-TiO2/Co ZnIn2S4 is approximately 500nm. The hydrogen production of 20-TiO2/Co ZnIn2S4 is also better than that of 20-TiO2/Co ZnIn2S4. The results obtained from the two characterizations are consistent.

23-   In Fig 10b discussion, “It also can be seen from figure 10b that the hydrogen production using non noble metal cocatalyst MoP (7.42 mmol•g-1) is twice more than that by using Pt cocatalyst (3.88 mmol•g-1)”. The data does not show twice more or doubled rate for any of the times studied.

Reply: Thank you very much for your comment. We have been checked in the manuscript and marked red. Please forgive our negligence.

It also can be seen from figure 10b that the hydrogen production by using non noble metal cocatalyst MoP (7.42 mmol·g-1) is approximately twice as much as that by using Pt cocatalyst (3.88 mmol·g-1).

24-  In Fig 10c discussion, the same material (MoP/a-TiO2/Co-ZnIn2S4) is being compared. Comparison with Pt/P25 should also be included.

Reply: Thank you very much for your comment. We have adjusted the position of sentences in the manuscript according to your suggestions

In addition, it can be observed from Figure 10b and Figure 10c that the hydrogen production capacity and hydrogen production rate of Pt/P25 at 2.5h is 6.43 mmol/g and 2.55 mmol·h-1·g-1 respectively.

25-  Fig 10d, stability should be discussed in terms of percentage drop or give a better understanding of the materials performance.

Reply: Thank you very much for your comment. We have been added in the manuscript and marked red.

As shown in Figure 10d, after using for three cycles, the hydrogen production only decreased by 5%, indicating that it has good cycle stability. But after 5 cycles, a slight decline in activity was observed (decline rate is 13.5%). This indicated a certain degree of catalytic stability. The degradation between cycles 3 and 4 (decline rate is 4.2%)was greater than that between cycles 1 and 2 (decline rate is 2.1%) due to the photocorrosion of ZnIn2S4.

26-  Line 392. What is the reference used for amorphous Fermi level of TiO2? Please include a reference for this information.

Reply: Thank you very much for your comment. We have been added in the manuscript.

[37] I. Mora-Seró, Bisquert J . Fermi Level of Surface States in TiO2 Nanoparticles[J]. Nano Letters, 2003, 3(7):945-949.

Reviewer 2 Report

In this manuscript, the authors synthesized the MoP/a-TiO2/Co-ZnIn2S4 flower-like composite and measured the performance of the composite in hydrogen production. I would suggest it to be accepted, provided to the following issues are addressed.

1)      In Figure 1, the authors mentioned different steps to synthesise the composites. The authors should add all the details and steps involved in this manuscript. In step 4, the authors mentioned the temperature etc.

2)      In Figure 5, the authors should index the peaks and add more information related to the crystalline size, lattice parameters, d spacing, etc.

3)      The author mentioned only one JCPDS card No for ZnIn2S4. The authors provide information related to the other compounds.

4)      In XPS data, why the cobalt peak was not visible properly? The authors should explain the reason. In the combined XPS graph, there are some extra peaks. What is the origin of these peaks

5)        In Figure 8, the authors measure the UV measurement, and the spectrum changed significantly following Co doping. The authors should add more details to justify this one.

6)      In Figure 9c, the authors measured the photocurrent. Did the authors check the on/off ratio, rise and decaying time and responsivity?

7)      Many spelling and formatting typos in this paper and the authors should check and revise them thoroughly.   

Many spelling and formatting typos in this paper, and the authors should check and revise them thoroughly.   

Author Response

Dear Editor and Reviewers:
Many thanks for your email and for giving us the revision opportunity on our manuscript entitled “Synthesis and hydrogen production performance of MoP/a-TiO2/Co-ZnIn2S4 flower-like composite photocatalysts” (Manuscript ID: molecules-2353011). Those comments are all valuable and very helpful for revising and improving our paper, as well as the important guiding significance to our researches. We tried our best to improve the manuscript and made some changes in the revised manuscript. We have listed changes at the end of this letter. And we would like to submit our corrected manuscript to “molecules”.
Comments and Suggestions for Authors
Reviewer2
Comments and Suggestions for Authors
In this manuscript, the authors synthesized the MoP/a-TiO2/Co-ZnIn2S4 flower-like composite and measured the performance of the composite in hydrogen production. I would suggest it to be accepted, provided to the following issues are addressed.
1) In Figure 1, the authors mentioned different steps to synthesise the composites. The authors should add all the details and steps involved in this manuscript. In step 4, the authors mentioned the temperature etc.
Reply: Thank you very much for your comment. We have made modifications to the diagram and completed the replacement, adding all the details. We have been added in the manuscript and marked red.
A mixture of 1 g Na2MoO4•2H2O and 10 g NaH2PO2 was ground in a mortar for 0.5 h until no crystal particles remained, and then transferred to a tubular furnace, under Ar protection at 400°C (heating rate 10°C/min), and calcined for 1 h to obtain MoP. 

2) In Figure 5, the authors should index the peaks and add more information related to the crystalline size, lattice parameters, d spacing, etc.
Reply: Thank you for pointing this out. We have researched the literature and supplemented the relevant parameters. We have been added in the manuscript and marked red.
ZnIn2S4 is a direct bandgap semiconductor with a layered structure. According to card NO. 48-1778, a=b=c=10.6,α=β=γ=90°),and trigonal (ICSD-JCPDS card NO. 49-1562, a=b=3.85, c=24.68, α=37.01°,β=90°,γ=120°), as shown in Figure 5. All polymorphs show certain photocatalytic performance under visible light, while the hexagonal ZnIn2S4 has better photocatalytic performance. The cubic ZnIn2S4 is a direct cubicspinel phase when the S atoms in the unit cell are ABC stacking[32].
[32] W. Yang, B. Liu, T. Fang, et al. Layered crystalline ZnIn2S4 nanosheets: CVD synthesis and photo-electrochemical properties[J]. Nanoscale, 2016, 8: 18197–18203.
3) The author mentioned only one JCPDS card No for ZnIn2S4. The authors provide information related to the other compounds.
Reply: Thank you very much for your suggestion, your suggestion is very pertinent. The shape of the ZnIn2S4 peaks did not change after TiO2 loading and no separate TiO2 peaks were identified, indicating that partly amorphous TiO2 was synthesized. The diffraction peaks also did not change after the addition of MoP, due to insufficient addition, the peak pattern is not displayed on the spectrum
4) In XPS data, why the cobalt peak was not visible properly? The authors should explain the reason. In the combined XPS graph, there are some extra peaks. What is the origin of these peaks?
Reply: Thank you for pointing this out. The content in the sample is very low, usually below 1%, and the resulting peak will be less obvious. In this study, the amount of Co added was around 0.5% (as can be seen from the amount added during the preparation process and the data in EDS). The inexplicable sources of photoelectron lines in XPS include impure or contaminated anode materials, and some X-rays come from impurities and trace elements.
5) In Figure 8, the authors measure the UV measurement, and the spectrum changed significantly following Co doping. The authors should add more details to justify this one.
Reply: Thank you very much for your comment. We have been added in the manuscript and marked red.
Figure 8b shows UV-vis diffuse reflectance spectra of ZnIn2S4, Co-ZnIn2S4, a-TiO2/Co-ZnIn2S4, and MoP/a-TiO2/Co-ZnIn2S4. Absorption edge of pure ZnIn2S4 synthesized in this study is 480 nm. As shown, the spectrum significantly changed after Co doping, reached 500 nm.

6) In Figure 9c, the authors measured the photocurrent. Did the authors check the on/off ratio, rise and decaying time and responsivity?
Reply: Thank you very much for your comment. We did fix the parameters you mentioned during the testing. During the test, the time for turning on and off the light was 20 seconds, and the rise and decay times of the obtained spectrogram were the same, and the response was approximately the same.
7) Many spelling and formatting typos in this paper and the authors should check and revise them thoroughly.
Reply: Thanks for your suggestion. We have reprocessed the language of the article and asked our school's foreign teachers to help us polish it.

Reviewer 3 Report

Upon reviewing your manuscript intitled: Synthesis and hydrogen production performance of MoP/a-TiO2/Co-ZnIn2S4 flower-like composite photocatalysts. I find your work interesting, but I do not believe it can be published in its current form. Therefore, some revisions must be done so that it may be published in this journal. I have the following comments and recommendations.

- The abstract is not attractive; the authors must provide some of the most important results of the characterization study.

-The introduction of relevant background and research progress was not comprehensive enough.

-Line 74, page 2. Use full name for XRD and XPS when first used.

- The doping can improve several material properties and the choice for doping depends on several factors. With respect to this, the authors do not justify the doping concentrations used in this system. Due to the importance for this system, a reasoned explanation must be attached as part of the motivation and justification of the work.

- Include the purity of the raw materials used.

- What is the pH value of the solution?

- Comments on morphology must be supported by bibliographical references because otherwise they are hypothetical.

-The rings shown in figure 4c correspond to which crystallographic planes?

-Structural parameters such as lattice constant, lattice strain and average crystallite size should be calculated and discussed in terms of doping

- Information on the deconvolution of XPS spectra must be provided. What program was used for the adjustment? what is the G/L ratio used in these analyses.

- What are the concentrations of Ti3 and Ti4. Use deconvolution to quantify this information.

Author Response

Dear Editor and Reviewers:
Many thanks for your email and for giving us the revision opportunity on our manuscript entitled “Synthesis and hydrogen production performance of MoP/a-TiO2/Co-ZnIn2S4 flower-like composite photocatalysts” (Manuscript ID: molecules-2353011). Those comments are all valuable and very helpful for revising and improving our paper, as well as the important guiding significance to our researches. We tried our best to improve the manuscript and made some changes in the revised manuscript. We have listed changes at the end of this letter. And we would like to submit our corrected manuscript to “molecules”.
Comments and Suggestions for Authors
Reviewer3
Comments and Suggestions for Authors
Upon reviewing your manuscript intitled: Synthesis and hydrogen production performance of MoP/a-TiO2/Co-ZnIn2S4 flower-like composite photocatalysts. I find your work interesting, but I do not believe it can be published in its current form. Therefore, some revisions must be done so that it may be published in this journal. I have the following comments and recommendations.
1. The abstract is not attractive; the authors must provide some of the most important results of the characterization study.
Reply: Thank you very much for your comment. We have added relevant data to the summary in the manuscript and marked red.
The absorption edge of MoP/a-TiO2/Co-ZnIn2S4 was widened from 480 nm to about 518 nm, and the specific surface area increased from 41.29 m2/g to 53.25 m2/g. The hydrogen production performance of this composite catalyst was investigated using a simulated light photocatalytic hydrogen production test system, and the rate of hydrogen production by MoP/a-TiO2/Co-ZnIn2S4 was found to be 2.96 mmol•h-1•g-1, which was three times than that of the pure ZnIn2S4 (0.98 mmol•h-1•g-1). After using for three cycles, the hydrogen production only decreased by 5%, indicating that it has good cycle stability.
2. The introduction of relevant background and research progress was not comprehensive enough.
Reply: Thank you very much for your comment. We have made relevant modifications to the introduction and highlighted it in red in the manuscript
The energy crisis is an ongoing global issue of increasing importance. Moreover, the rapid development of industrialization around the world has led to severe energy and environmental pressures [1]. Thus, there is an increased emphasis on research worldwide to successfully address the global energy crisis and to create new sustainable sources of energy [2]. The capture and conversion of solar energy by the photocatalytic splitting of water offers a promising strategy for converting inexhaustible solar energy into hydrogen (H2) energy [3]. However, there are currently two main constraints that limit the large-scale application of hydrogen. One, the large-scale green synthesis of hydrogen is a significant challenge. Two, the storage and transport of hydrogen is also difficult [4]. Hydrogen's shortcomings are partly explained by high infrastructure costs for production, storage, and distribution; owing to its low energy density per volume, explosive characteristics, and ability to cause embrittlement in metals such as steel[34]. Many methods have been investigated for the production of hydrogen. The photocatalytic decomposition of water for hydrogen production is one of the simplest, most environmentally friendly, and low-cost methods for producing hydrogen. Therefore, this method has attracted extensive research attention [5]. In particular, the production of hydrogen via the solar photolysis of water is gaining increasing attention due to its potential for solving the global energy crisis and mitigating environmental pollution problems [6,33]. Photogenerated charge carriers can be excited from photocatalysts under sunlight, and after the photogenerated electrons migrate to the surface of semiconductors, H+ in water receive electrons that are reduced to H2. The holes left behind are combined with sacrificial agents in the system and used to achieve continuous H2 production.
In photocatalytic systems, the mobility of photogenerated carriers is an important factor affecting photocatalytic efficiency, with a fast migration rate and high separation efficiency positively contributing to the photocatalytic reaction [7]. The electrostatic potential of ZnIn2S4 with a hexagonal laminar structure is uniformly distributed within the plane, and the small potential of this material is well conducive to carrier migration[8]. Moreover, the positive charges are densely distributed in the indium sulfide tetrahedra and octahedra within the cell, while the negative charges are concentrated in the zinc indium tetrahedral[9]. Therefore, photogenerated electrons are easily transferred to the indium sulfide polyhedra, while the photogenerated holes more easily migrate to the zinc indium tetrahedra, which improves the separation efficiency of photogenerated carriers [10]. Furthermore, the band gap of ZnIn2S4 is 2.3~2.5 eV. The energy band of ZnIn2S4 is narrow, which is also conducive to the generation of photogenerated carriers. ZnIn2S4 is therefore an ideal photocatalytic material with broad application prospects [11].
In 2003, Lei [12] et al. synthesized ZnIn2S4 by a hydrothermal method and used this materialas an effective visible-light-driven hydrogen precipitation photocatalyst for the first time. Guoʹs group [13] synthesized ZnIn2S4 microspheres by a hydrothermal/solvothermal process and explored their visible-light-driven photocatalytic hydrogen production performance. Their findings showed that these microsphere catalysts had a good potential for producing photocatalytic hydrogen from water when exposed to visible light[14].
However, pure ZnIn2S4 photocatalysts still suffer from low visible light utilization and low photocatalytic activity [15]. Moreover, the photocatalytic activity of ZnIn2S4 semiconductors is affected to some extent by their limited photogenerated electron and hole separation efficiency under visible light irradiation and low photogenerated carrier mobility [16]. Therefore, Yuan Wenhui [17]et al. prepared a series of Co-doped ZnIn2S4 photocatalysts using a solvothermal synthesis method. The successful incorporation of Co into the ZnIn2S4 lattice was confirmed by X-ray diffraction (XRD) and X-ray photoelectron spectroscopy (XPS). With increasing Co concentration, the absorption edge of the samples caused red shift, but the Co also gradually disrupted the ZnIn2S4 morphology. Their photocatalytic results showed that Co2+ doping significantly improved the photocatalytic activity of ZnIn2S4. The optimum Co doping amount of 0.3wt% for the ZnIn2S4 photocatalyst led to the highest photocatalytic activity [18]. Therefore, in this work, a doping amount of 0.3% was chosen to preserve the petal-like morphology and enhance the specific surface area of ZnIn2S4 while also improving its hydrogen production performance and utilization of sunlight [19,35].
TiO2 has been widely investigated as a semiconductor photocatalyst material due to its many advantages, such as high stability and high photosensitivity. Therefore, TiO2-based metal oxide photocatalyst are widely used in many practical applications [20]. However, TiO2 particles easily agglomerate, have a low adsorption capacity for organic matter, and exhibit low solar energy utilization [21]. These factors limit the photocatalytic efficiency of TiO2 and seriously affect its application in practical production [22]. The focus of photocatalytic research has therefore shifted from the improvement of traditional TiO2 performance to the investigation of other catalysts with better performance in the visible light range. Amorphous TiO2 is an important category of TiO2 materials that exhibits the common "short-range order, long-range disorder"[23] structural feature seen in amorphous materials. Amorphous semiconductors have a large number of suspended bonds. Therefore, the energy band structures of amorphous materials exhibit a gap band between the valence band and the conduction band [24]. Amorphous TiO2 with a lower band gap width can be obtained by modifying its electronic structure. This reduces the energy intensity required for electrons to transfer from the valence band to the conduction band [25]. Therefore, visible light irradiation can be used to activate these materials, improving their photocatalytic activity [26]. Zywitzki D reported amorphous titania-based photocatalysts were synthesized using a facile, UV-light mediated method and evaluated as photocatalysts for hydrogen evolution from water/methanol mixtures. The resulting amorphous materials exhibit an overall higher hydrogen evolution rate (1.09 mmol•h-1•g-1 )compared to a crystalline TiO2 reference (P25 0.80 mmol•h-1•g-1) on a molar basis of the photocatalyst due to their highly porous structure and high surface area[36].
The photocatalytic activity of a photocatalyst is determined by its light absorption capacity as well as its electron-hole transfer and separation efficiency [27]. These factors are related to the catalyst surface properties, which play an important role in photocatalytic processes. For instance, the loading of co-catalysts on a photocatalyst surface to provide hydrogen production sites has been commonly reported in the literature [28]. Some common co-catalysts include alumina and potassium oxide. MoP is commonly used as an efficient catalyst for hydrodesulfurization(HDS) and hydrodenitrogenation (HDN) reactions [29]. Depending on the reversibility of hydrogen bonding to the catalyst, some catalysts used for HDS reactions are also useful for HER reactions because of the similar pathways and mechanisms of hydrogen production and hydrogenation as well as their low Tafel slope and low over potential. For example, Chen [30] et al. impregnated precursors on sponges to obtain MoP with a large specific surface area and enhanced photocatalytic activity. MoP cannot be directly used as a photocatalyst, but it can be used as an efficient hydrogen precipitation co-catalyst. Du et al. used MoP as a highly active co-catalyst on CdS nanorods for the first time, which significantly improved the photocatalytic activity of their CdS catalyst[31]. Thus, MoP is an efficient co-catalyst for hydrogen precipitation.
At present, the utilization of solar energy by metal oxide photocatalyst for hydrogen production has mainly focused on the UV wavelength range. Furthermore, most research is based on TiO2 semiconductor photocatalytic materials. The majority of the wavelengths that make up solar energy, though, do not fall inside the visible spectrum. ZnIn2S4 shows promise as a visible-light-responsible ternary metal-sulfur compound photocatalyst, but its performance still needs to be improved. Therefore, in this work, ZnIn2S4 materials were prepared and modified, as shown in Figure 1: (1) Petal-shaped ZnIn2S4 catalysts were produced, their morphology was studied, and their photocatalytic performance was investigated. (2) Co-ZnIn2S4 was prepared by Co doping and ultrasonic exfoliation to broaden the absorption band edge and retain the petal-shaped morphology of the catalyst. (3) An a-TiO2/Co-ZnIn2S4 composite photocatalyst was successfully prepared by coating amorphous TiO2 on the Co-ZnIn2S4 surface, and the effect of loading different amounts of TiO2 on the photocatalytic performance was investigated. At the same time, a TiO2 and Co-ZnIn2S4 heterojunction was constructed, which led to the red-shift of the absorption band, enhanced light absorption properties, and a reduction in photogenerated electron-hole recombination. (4) Finally, MoP was loaded on the a-TiO2/Co-ZnIn2S4catalyst as a co-catalyst, which enhanced the light absorption intensity and provided reaction sites to promote the overall efficiency of catalytic hydrogen production. Therefore, MoP/a-TiO2/Co-ZnIn2S4 flower-like composite photocatalysts with good photocatalytic hydrogen production activity and stability were prepared. This catalyst uses Co-ZnIn2S4 as the main body for photo generated electron excitation, and amorphous a-TiO2 is combined with it to improve the efficiency of electron hole separation. Finally, MoP is used as a co catalyst to provide hydrogen production sites, thus achieving efficient hydrogen production.
3. Line 74, page 2. Use full name for XRD and XPS when first used.
Reply: Thank you very much for your comment. We have made modifications in the manuscript and marked red.
The successful incorporation of Co into the ZnIn2S4 lattice was confirmed by X-ray diffraction (XRD) and X-ray photoelectron spectroscopy (XPS).
4.The doping can improve several material properties and the choice for doping depends on several factors. With respect to this, the authors do not justify the doping concentrations used in this system. Due to the importance for this system, a reasoned explanation must be attached as part of the motivation and justification of the work. Include the purity of the raw materials used. What is the pH value of the solution?
Reply: Thank you very much for your comment. All reagents used are analytical pure (A.R. 99.7%). In order to hydrolyze Co2+, the pH value of the solution was adjusted to 11.3.
Therefore, Yuan Wenhui [17]et al. prepared a series of Co-doped ZnIn2S4 photocatalysts using a solvothermal synthesis method. The successful incorporation of Co into the ZnIn2S4 lattice was confirmed by X-ray diffraction (XRD) and X-ray photoelectron spectroscopy (XPS). With increasing Co concentration, the absorption edge of the samples caused red shift, but the Co also gradually disrupted the ZnIn2S4 morphology. Their photocatalytic results showed that Co2+ doping significantly improved the photocatalytic activity of ZnIn2S4. The optimum Co doping amount of 0.3wt% for the ZnIn2S4 photocatalyst led to the highest photocatalytic activity [18]. Therefore, in this work, a doping amount of 0.3% was chosen to preserve the petal-like morphology and enhance the specific surface area of ZnIn2S4 while also improving its hydrogen production performance and utilization of sunlight [19,35].
According to relevant literature research (17,18,19,35), we have selected a doping amount of 0.3%.
0.136 g zinc chloride, 0.586 g indium chloride, and 0.301 g thioacetamide were weighed and added to 80 mL ethylene glycol. This mixture was stirred and centrifugally sonicated to dissolve the solid compounds. 
According to the amount of reagent used, Co2+ addition is 0.3%. Through EDS analysis of surface elements, it can be seen that the Co element content in the selected area is 0.5%.
 In S Zn Mo P Ti O Co
Wt% 53.2 26.0 10.2 2.5 0.3 1.1 6.2 0.5

5. Comments on morphology must be supported by bibliographical references because otherwise they are hypothetical.
Reply: Thank you very much for your comment. There is relevant literature support for flower morphology, and we have added relevant references.
Therefore, in this work, a doping amount of 0.3% was chosen to preserve the petal-like morphology and enhance the specific surface area of ZnIn2S4 while also improving its hydrogen production performance and utilization of sunlight [19,35].
According to the literature: As can be seen, the pure ZIS is self-organized into a uniquehierarchical peony-like spherical structure (Fig. 1a and e).
[35] X. Yang, Q. Li, K. L. Lv, et al. Superiority of graphene over carbon analogs for enhanced photocatalytic H2-production activity of ZnIn2S4[J]. Appl. Catal. B. 2017: 344-352.
6. The rings shown in figure 4c correspond to which crystallographic planes?
Reply: Thank you very much for your comment. After being processed by the relevant processing software of transmission electron microscopy and compared to the PDF card in the positive space, it can be concluded that this diffraction ring is a 102 crystal plane.
7. Structural parameters such as lattice constant, lattice strain and average crystallite size should be calculated and discussed in terms of doping.
Reply: Thank you very much for your comment. DFT calculation can obtain structural parameters such as lattice constant, lattice strain, and average grain size. But the related software is quite difficult, and our team does not have the relevant resources. I feel very sorry for the use of related software. If entrusted to a testing company, the calculation cycle is two months and the related fees are very expensive. We will strive to introduce a team of relevant teachers in the future, and strive to include the calculation of relevant crystal parameters in future research. Please forgive us for not being able to provide the data obtained by the relevant software in this study.
8. Information on the deconvolution of XPS spectra must be provided. What program was used for the adjustment? what is the G/L ratio used in these analyses. What are the concentrations of Ti3+ and Ti4+. Use deconvolution to quantify this information.
Reply: Thank you very much for your comment. Avantage is a software used for peak processing of data. The L/G ratio is 20% in these analyses. Useing deconvolution the area of two peaks can be obtained, and combined with a Ti element sensitivity factor of 2.001, the ion concentration can be obtained. However, only the relative content of ions, the ratio of ion concentrations, can be determined. The ratio of peak area obtained from deconvolution in this data graph is approximately 8:9, That is to say, the ratio of Ti3+ and Ti4+concentration is 8:9.

Reviewer 4 Report

Molecules

Synthesis and hydrogen production performance of MoP/a- 2TiO2/Co-ZnIn2S4 flower-like composite photocatalysts; authors: K. Wu, Y. Shang, H. Li, P. Wu, S. Li, H. Ye, F. Jian, J. Zhu, D. Yang, B. Li, X. Wang

This manuscript can be published with the following observations:

- I think that the term “semiconductor photocatalysts” must be avoided, metal oxide photocatalyst can be used;

- The sentences from lines 33-35; 49-52; 65-67; 117-118 must be revised;

- line 57 band gap not bandage;

- The text from the lines 313-328 must be revised and a new discussion must be made. In Fig. 8a, 8b on the oy axis give the reflectance, not intensity. In line 322 absorption edge not ”absorption band edge”. The band gap energy results from Tauc plots (Fig. 8c). Revised the discussion of the obtained data;

- Chapter 3.2 must be revised. Fig. 9a should be shown the high frequency semicircles, but they are not present even in inset Fig. 9a. Fig. 9c was not correctly discussed, the semicircles can not evinced in this figure. Fig. 9c does not contain its legend;

- The authors must differentiate in text the discussion on the results relating to TiO2/Co-ZnIn2S4 composite TiO2 and Co-ZnIn2S4heterojunction, according to the above mentioned in introduction;

- The references must be written according to the journal requirements;

- The  English can be improved. The typos must be checked and corrected in text.

The  English can be improved. The typos must be checked and corrected in text.

Author Response

Many thanks for your email and for giving us the revision opportunity on our manuscript entitled “Synthesis and hydrogen production performance of MoP/a-TiO2/Co-ZnIn2S4 flower-like composite photocatalysts” (Manuscript ID: molecules-2353011). Those comments are all valuable and very helpful for revising and improving our paper, as well as the important guiding significance to our researches. We tried our best to improve the manuscript and made some changes in the revised manuscript. We have listed changes at the end of this letter. And we would like to submit our corrected manuscript to “molecules”.

Comments and Suggestions for Authors

Reviewer4

In the manuscript presented for review, Wang et al. described the synthesis of semiconductors and their application in the photolytic hydrogen production process. The work has been prepared carefully in terms of content, and my critical remarks concern the editorial aspect.

  • On pages 2 and 3, authors should pay more attention to subscripts in structural formulas (they just don't exist...).

Reply: Thank you very much for your comment. It should be a problem with the editing system of the journal, because the corner marks were correct in my original manuscript, and there was indeed an issue with the two pages after uploading to the system. I hope that similar system issues will not occur in the revised manuscript.

  • Authors should carefully follow the entire text and eliminate places where spaces are not used.

Reply: Thank you very much for your comment. We will carefully examine this issue, and please forgive our negligence.

  • Ethyl alcohol doesn't have the formula CH3COOH, it's rather acetic acid, Page 4, line 144.

Reply: Thank you very much for your comment. We have made modifications in the manuscript and marked red. Please forgive our negligence. Ethyl alcohol (CH3CH2OH, AR)

  • Page 6, line 245 - The word 'catalysts' is in larger font.

Reply: Thank you very much for your comment. We have been added in the manuscript and marked red. Please forgive our negligence.

Distribution of elements in the MoP/a-TiO2/Co-ZnIn2S4 composite catalyst

  • There is an inconsistency in the spelling of journal names in the list of references. The authors sometimes use the full names of the journals, and sometimes their abbreviations. This should be unified.

Reply: Thanks for your suggestion. We have carefully examined this issue and have adjusted it all to a unified format.

Having considered the above editorial remarks by the authors, I can recommend this manuscript for publication in Molecules.

Reviewer 5 Report

In the manuscript presented for review, Wang et al. described the synthesis of semiconductors and their application in the photolytic hydrogen production process. The work has been prepared carefully in terms of content, and my critical remarks concern the editorial aspect.

1) On pages 2 and 3, authors should pay more attention to subscripts in structural formulas (they just don't exist...).

2) Authors should carefully follow the entire text and eliminate places where spaces are not used.

3) Ethyl alcohol doesn't have the formula CH3COOH, it's rather acetic acid, Page 4, line 144.

4) Page 6, line 245 - The word 'catalysts' is in larger font.

5) There is an inconsistency in the spelling of journal names in the list of references. The authors sometimes use the full names of the journals, and sometimes their abbreviations. This should be unified.

Having considered the above editorial remarks by the authors, I can recommend this manuscript for publication in Molecules.

Author Response

Dear Editor and Reviewers:
Many thanks for your email and for giving us the revision opportunity on our manuscript entitled “Synthesis and hydrogen production performance of MoP/a-TiO2/Co-ZnIn2S4 flower-like composite photocatalysts” (Manuscript ID: molecules-2353011). Those comments are all valuable and very helpful for revising and improving our paper, as well as the important guiding significance to our researches. We tried our best to improve the manuscript and made some changes in the revised manuscript. We have listed changes at the end of this letter. And we would like to submit our corrected manuscript to “molecules”.
Comments and Suggestions for Authors
Reviewer5
Synthesis and hydrogen production performance of MoP/a- 2TiO2/Co-ZnIn2S4 flower-like
composite photocatalysts; authors: K. Wu, Y. Shang, H. Li, P. Wu, S. Li, H. Ye, F. Jian, J. Zhu, D.
Yang, B. Li, X. Wang
This manuscript can be published with the following observations:
1. I think that the term “semiconductor photocatalysts” must be avoided, metal oxide photocatalyst canbe used;
Reply:Thank you very much for your comment. We have been added in the manuscript and marked red. 
Therefore, TiO2-based metal oxide photocatalyst are widely used in many practical applications.
At present, the utilization of solar energy by metal oxide photocatalyst for hydrogen production has mainly focused on the UV wavelength range.
2. The sentences from lines 33-35; 49-52; 65-67; 117-118 must be revised;
Reply: Thank you very much for your comment. We have been added in the manuscript and marked red.
Lines 33-35:Thus, there is an increased emphasis on research worldwide to successfully address the global energy crisis and to create new sustainable sources of energy[2].
49-52:The electrostatic potential of ZnIn2S4 with a hexagonal laminar structure is uniformly distributed within the plane, and the small potential of this material is well conducive to carrier migration[8]. Moreover, the positive charges are densely distributed in the indium sulfide tetrahedra and octahedra within the cell, while the negative charges are concentrated in the zinc indium tetrahedra.
65-67:Their findings showed that these microsphere catalysts had a good potential for producing photocatalytic hydrogen from water when exposed to visible light[14].
117-118:The majority of the wavelengths that make up solar energy, though, do not fall inside the visible spectrum.
3. line 57 band gap not bandage;
Reply: Thank you very much for your comment. We have been checked in the manuscript and marked red. Please forgive our negligence.
Furthermore, the band gap of ZnIn2S4 is 2.3~2.5 eV, and The energy band of ZnIn2S4 is narrow, which is also conducive to the generation of photogenerated carriers[11].
4. The text from the lines 313-328 must be revised and a new discussion must be made. In Fig. 8a, 8b on the oy axis give the reflectance, not intensity. In line 322 absorption edge not ”absorption band edge”. The band gap energy results from Tauc plots (Fig. 8c). Revised the discussion of the obtained data;
Reply: Thank you very much for your comment. We have been added in the manuscript and marked red. At the same time, we have replaced the images again.
Figure 8b shows UV-vis diffuse reflectance spectra of ZnIn2S4, Co-ZnIn2S4, a-TiO2/Co-ZnIn2S4, and MoP/a-TiO2/Co-ZnIn2S4. Absorption edge of pure ZnIn2S4 synthesized in this study is 480 nm. As shown, the spectrum significantly changed after Co doping, reached 500 nm. However, almost the same of absorption was demonstrated after the addition of amorphous TiO2. The absorption edge of MoP/a-TiO2/Co-ZnIn2S4 was widened to about 518 nm, indicating that the MoP was photocatalyzed by the load. The band gap energy of ZnIn2S4, Co-ZnIn2S4, a-TiO2/Co-ZnIn2S4, and MoP/a-TiO2/Co-ZnIn2S4 were calculated using the curves shown in Figure 8c. The band gap energy of MoP/a-TiO2/Co-ZnIn2S4 was 2.7 eV. This analysis demonstrated the broader light absorption range and enhanced photocatalytic activity of the MoP/a-TiO2/Co-ZnIn2S4 composite photocatalyst.

5. Chapter 3.2 must be revised. Fig. 9a should be shown the high frequency semicircles, but they are not present even in inset Fig. 9a. Fig. 9c was not correctly discussed, the semicircles can not evinced in this figure. Fig. 9c does not contain its legend;
Reply: Thank you very much for your comment. Please forgive our negligence. We retested the EIS and replaced the spectrogram. 

Figure 9 (a) Photocurrent response curves of Co-ZnIn2S4, a-TiO2/Co-ZnIn2S4, and MoP/a-TiO2/Co-ZnIn2S4 (b) linear scanning voltammograms and photocurrent response curves of ZnIn2S4, Co-ZnIn2S4, a-TiO2/Co-ZnIn2S4, and MoP/a-TiO2/Co-ZnIn2S4;(c) EIS curves of Co-ZnIn2S4, a-TiO2/Co-ZnIn2S4, and MoP/a-TiO2/Co-ZnIn2S4;
6. The authors must differentiate in text the discussion on the results relating to TiO2/Co-ZnIn2S4 composite TiO2 and Co-ZnIn2S4 heterojunction, according to the above mentioned in introduction.
Reply: Thank you very much for your comment. We have supplemented relevant data on P25 hydrogen production and hydrogen production rate. We have been added in the manuscript and marked red. At the same time, we have replaced the images again.
In addition, it can be observed from Figure 10b and Figure 10c that the hydrogen production capacity and hydrogen production rate of Pt/P25 at 2.5h is 6.43 mmol/g and 2.55 mmol•h-1•g-1 respectively.
7. The references must be written according to the journal requirements;
Reply: Thanks for your suggestion. We have carefully examined this issue and have adjusted it all to a unified format.
8. The English can be improved. The typos must be checked and corrected in text.
Reply: Thanks for your suggestion. We have reprocessed the language of the article and asked our school's foreign teachers to help us polish it.

Round 2

Reviewer 2 Report

 Accept in present form.

Minor editing of English language required

Reviewer 3 Report

This version can be accepted for publication.